# Ultra-low friction graphene oxide in the Atotsugawa Fault System

Tomoya Shimada [1] ✉, Hiroyuki Nagahama[1], Jun Muto [1], Norihiro Nakamura [2], Sando Sawa [1] & Hiroaki Ohfuji[3]

The active Atotsugawa Fault System in central Japan exhibits low levels of seismicity and creep to depths of 7–8 km. Low-friction graphite (friction coefficient, $\mu \sim 0.1$) in fault rocks has previously been proposed as a mechanism for reducing fault strength and contributing to aseismic creep. Here, we use state-of-the-art Raman spectroscopy, X-ray photoelectron spectroscopy (XPS) and transmission electron microscopy (TEM) to document for the first time the discovery of graphene oxide in natural fault gouge, a material characterized by an ultra-low coefficient of friction ($\mu \sim 0.01$) and possible superlubricity. Graphene oxide is concentrated within microcracks in fault gouge and occurs as single-layer sheets with particle sizes of 3–10 nm. Analysis of hydroxyl groups and the degree of oxidation indicate that the graphene oxide has the most effective chemical composition for fault weakening. We suggest that graphene oxide in fault rocks could dramatically reduce fault frictional strength and promote fault creep.

The Atotsugawa Fault System is one of the most active fault systems in central Japan[1–3]. It consists of a network of aligned (fault strikes ~N60°E), subvertical strike-slip faults ~60 km long, including the Atotsugawa, Mozumi–Sukenobe, and Ushikubi faults (Fig. 1a; refs. 3,4). Linked rupture of these faults generated the 1858 $M$ 7.0 Hietsu earthquake (Fig. 1a; refs. 5–8). Dense seismological observations show that a low-seismicity region along the fault system extends to depths of 7–8 km.[4,9,10] This region is most pronounced in the central part of the fault system and decreases in thickness toward both ends (Fig. 1b), suggesting pronounced localization of hypocenters on the fault plane. Geodetic observations have suggested that this low seismicity distribution is linked to fault creep[1]. In particular, electronic distance measurement (EDM) observations for the period 1981–1999 indicate creep of ~1.5 mm/yr along the central part of the fault system (Fig. 1b; refs. 11,12). Instead, Global Positioning System (GPS) observations for the period 1998–2006 indicate a locked state, suggesting possible time-dependent fault creep behavior[2]. In addition, Interferometric Synthetic Aperture Radar (InSAR) and Global Navigation Satellite System (GNSS) analyses (2007–2010) show that the interseismic strain-rate gradient around the fault varies over time[13], which may reflect temporal changes in seismicity along the central region associated with stress field variations in the seismogenic zone[14].

The presence of graphite-bearing fault rocks has previously been interpreted as a possible cause of low seismicity and fault creep along the Atotsugawa Fault System[15]. Byerlee's law indicates that the friction coefficient of typical rocks and rock-forming minerals is between 0.6 and 0.85 (ref. 16). Clay minerals have lower friction coefficients ($\mu = 0.2$–0.8) and have been suggested to cause aseismic fault slip[16,17]. However, graphite, a well-known solid lubricant consisting of stacked layers of graphene[18,19], has an even lower coefficient of friction (~ 0.1) than the clay minerals[16,17,20,21]. Friction experiments show that graphite, which is stable over a wide range of geologically relevant pressure and temperature conditions, exhibits low friction over a range of slip rates (Fig. 1c; refs. 20,21). When mixed with quartz, even small amounts of graphite (~ 10 vol%) can significantly reduce frictional strength[21]. Fault gouge from the Ushikubi Fault in the Atotsugawa Fault System contains up to about 12 wt% graphite[15], enough to significantly reduce the frictional strength.

Shear during fault slip can produce low friction graphene[22,23]. It is well known that single layers of graphene can be produced by peeling off adhesive tape from the surface of graphite[24,25]. Graphene changes its

[1]Department of Earth Science, Graduate School of Science, Tohoku University, Sendai, Japan. [2]Office of Higher Education Development, Tohoku Gakuin University, Sendai, Japan. [3]Department of Earth and Planetary Science, The University of Tokyo, Tokyo, Japan. ✉e-mail: tomoya.shimada.s2@dc.tohoku.ac.jp

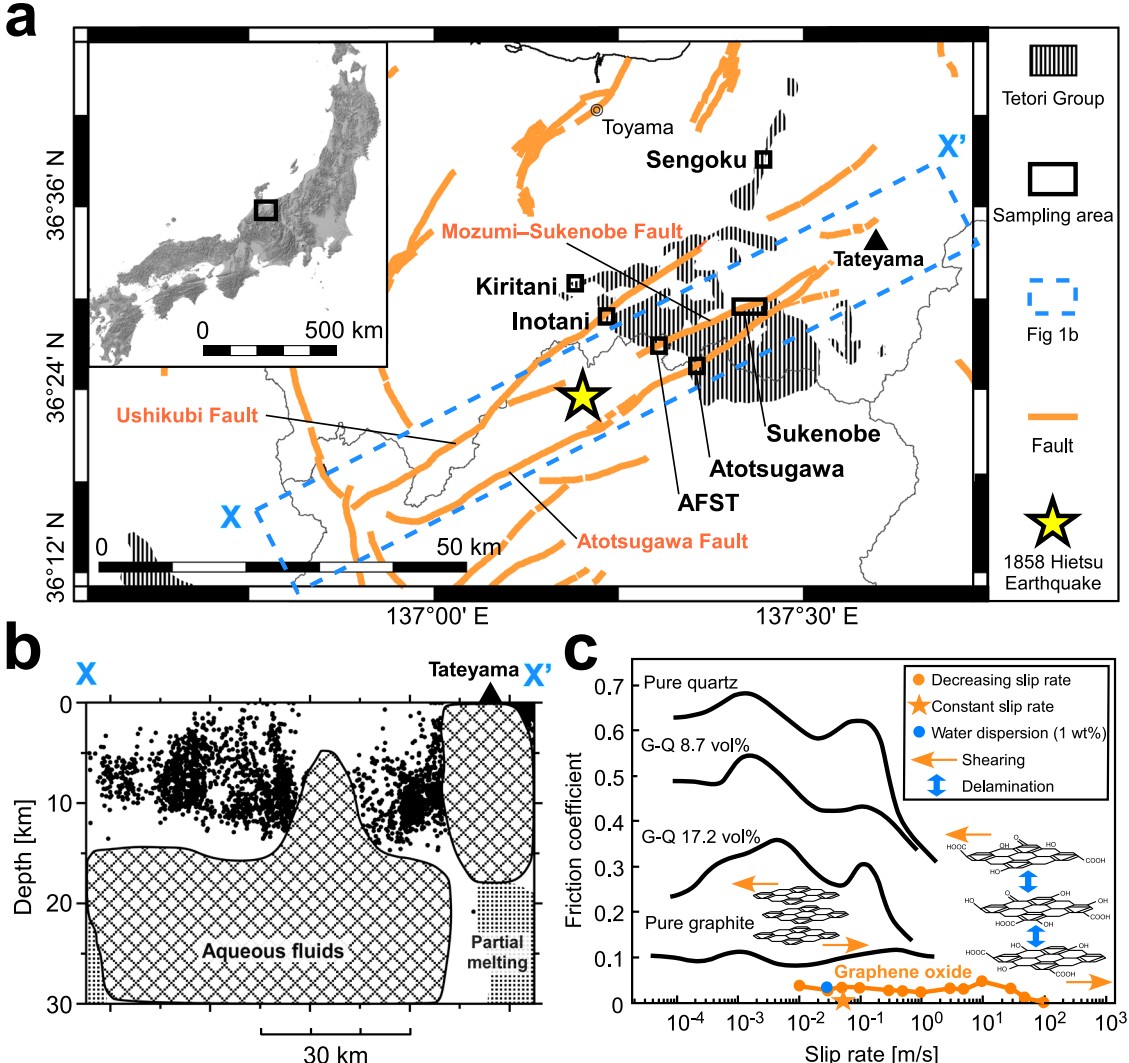

**Fig. 1 | Atotsugawa Fault System. a** Summary geological map of the Atotsugawa Fault System, modified after the Visualization System for Subsurface Structures (https://gbank.gsj.jp/activefault/index_e_gmap.html)[70]. Hatched areas = Tetori Group; orange bold lines = faults; black rectangles = study areas; star = hypocenter of the 1858 Hietsu earthquake[6,7]; AFST = Active Fault Survey Tunnel through the Mozumi–Sukenobe Fault. **b** Depth section of hypocenters from 1998 to 2025 in the area outlined with a blue dashed line between X and X' in this figure (**a**) (catalog from the Seismological Bulletin, Japan Meteorological Agency, https://www.data. jma.go.jp/eqev/data/bulletin/hypo.html). Seismic tomography suggests the presence of aqueous fluids and partial melting in the deep crust[9]. **c** Frictional properties of graphite and graphene oxide[21–23]. G-Q represents mixtures of graphite and quartz with different volume percentages of graphite. Orange and blue arrows represent shearing along the basal plane and interlayer delamination, respectively.

properties depending on morphology and redox conditions[22–31], and it has attracted interest from various scientific fields and industries[22–31]. For example, graphene oxide, which is formed by oxidation and formation of oxygen-bearing functional groups and defects, has an extremely low friction coefficient of ~0.01 (Fig. 1c; refs. 22,23). This is an order-of-magnitude lower than graphite ($\mu = 0.1$)[20,21] and several orders of magnitude lower than typical Byerlee rocks and minerals[17]. If graphene oxide is produced from carbonaceous material (CM) during fault slip, the implications for fault frictional behavior could be significant. Although it is unclear whether single-layer graphene is stable in rocks over geological timescales, previous studies have reported the presence of graphene and graphene-like materials from specific settings. They typically occur as an incompletely exfoliated graphene–graphite phase, or as aggregates of staked graphene in deposits of graphite and shungite formed through redox reactions in hydrothermal fluids[32,33]. Despite the exceptional physicochemical properties of graphene, it has received relatively little attention in a geological context, particularly in seismology and structural geology[34].

Although conventional methods of Raman spectroscopy can identify graphite from the G-band at ~1600 cm$^{-1}$ in Raman spectra[15,35], it has been a challenge to correctly identify and distinguish other types of CM in faults. Recent improvements in Raman spectroscopy allow better classification of different types of CM, including graphene, by decomposing the Raman spectra through analysis of positions, intensities, and the full-width-at-half-maximum (FWHM). In addition, X-ray photoelectron spectroscopy (XPS) can be used to quantitatively analyze the elemental composition and chemical bonding states of CM (e.g., carbon bonding, hydroxyl, and carboxyl groups)[30,31,36]. Although XPS is widely used in materials science and chemistry, it has rarely been applied to geological samples[36]. Furthermore, transmission electron microscopy (TEM) allows direct observation of the stacking structure of graphene and determination of the number of layers. Here, we combine state-of-the-art Raman spectroscopy, XPS, and TEM to document for the first time the presence of ultra-low friction graphene oxide in an active fault and to study its chemical bonding state and spatial distribution in fault rocks. The use of cutting-edge

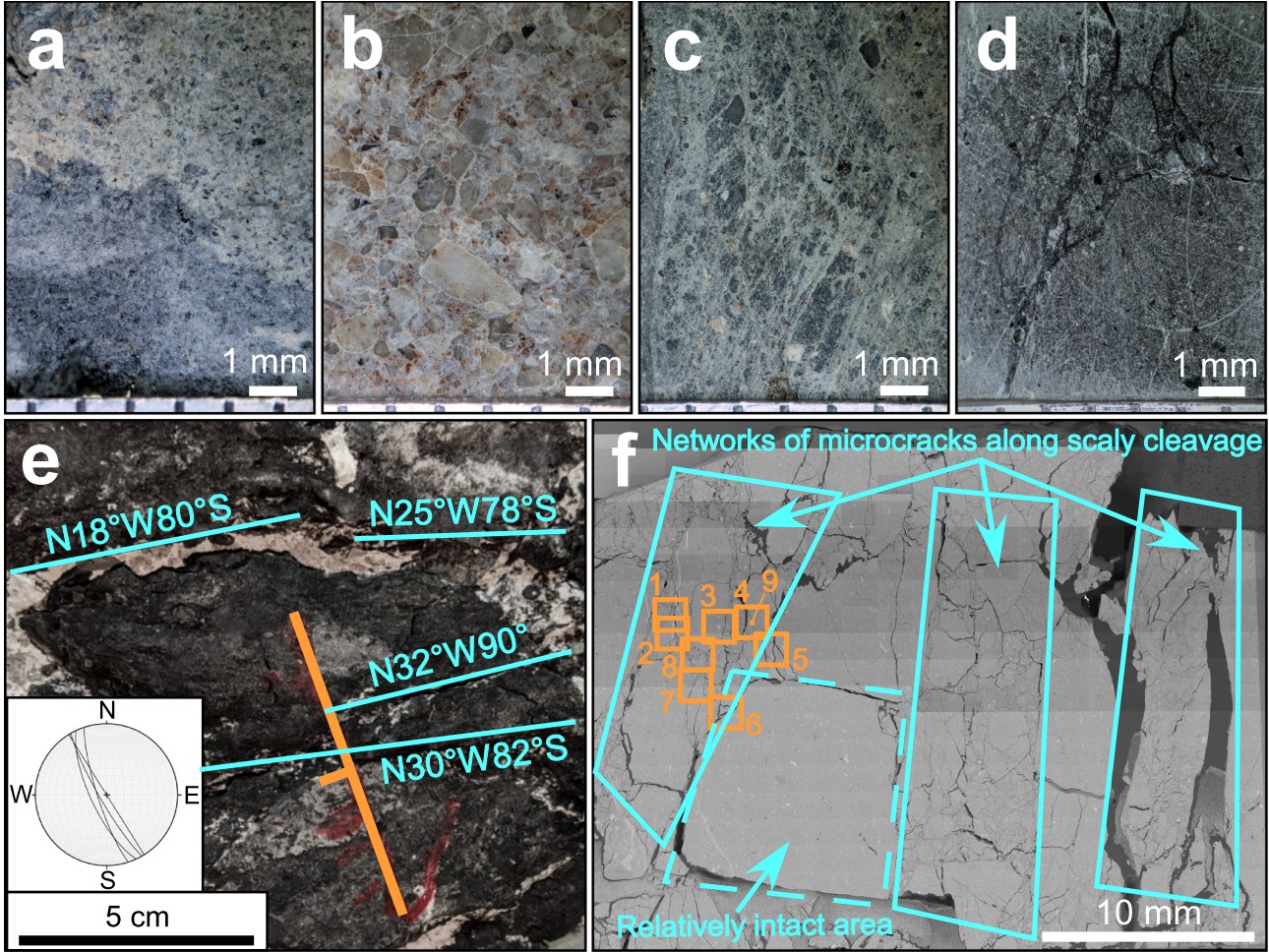

**Fig. 2 | Samples in which graphene oxide–like CM was identified. a–d** Optical microscope images. **a** Fault gouge from the Atotsugawa area (sample 1141). **b** Tetori Group from the Inotani area (sample 1506). **c** Fault gouge from the Sukenobe area (sample 1236). **d** Fault gouge from AFST (sample AFST). **e, f** Images of fault gouge from the AFST. **e** The gouge contains a scaly cleavage defined by color banding and a preferred orientation of gouge particles. Blue lines show orientations of representative cleavage surfaces and the inset shows a lower hemisphere stereographic projection of the cleavage surfaces. The orange symbol indicates the recorded strike and dip (N40°E, 76°S) of the oriented sample. **f** Backscattered electron SEM image of the XZ surface from the AFST gouge sample. The blue polygons show regions where networks of microcracks are visible around the edges of gouge particles and subparallel to scaly cleavage surfaces. The dashed blue polygon indicates a relatively intact area of gouge. The orange squares show the regions (labelled 1 to 9) used for XPS line analysis.

spectroscopic methods provides new insights into the potential effects of graphene oxide on the frictional and seismological properties of faults. We suggest that graphene oxide can significantly reduce fault frictional strength, potentially leading to fault creep (aseismic slip) and low seismicity along the Atotsugawa Fault System.

## Results

### Geological setting and sample observations

We collected samples of fault gouge and surrounding host rocks from six study areas along the Atotsugawa Fault and the Mozumi–Sukenobe Fault, which belong to the Atotsugawa Fault System in Gifu and Toyama Prefectures (Figs. 1a and S1–8 and Table S1). The host rocks belong to the Inotani alternation member (Nakanomatanokkoshi sandstone member) of the Nagatou-gawa Formation (Itoshiro subgroup), which is part of the Tetori Group[37–40]. The Tetori Group is widely distributed in central Japan (Fig. 1a) and comprises sedimentary rocks that were deposited from the Upper Jurassic to the Early Cretaceous[15,39–41]. Traditionally, the Tetori Group has been divided into lower units consisting of Jurassic ammonite-bearing marine formations, and middle and upper units consisting mainly of Early Cretaceous non-marine clastic formations[42]. In the following paragraphs, we describe the locations of the samples used in the study.

In the Atotsugawa Fault (Fig. 1a), we collected fault gouge (sample 1141) and sandstone host rock (sample 1124) from the central segment of the fault. The gouge in the Atotsugawa area (sample 1141; Fig. 2a) comprises distinct black and gray layers. The black gouge is composed of clay- to silt-sized particles, while the gray gouge contains abundant black CM grains ranging from silt to fine-sand in size. Raman spectroscopy of the gouge (sample 1141) identified graphene oxide–like CM, including graphene oxide and/or its stacks, for which the thickness and stacking structure cannot be constrained.

In the Mozumi–Sukenobe Fault (Fig. 1a), we collected fault gouge samples from a surface outcrop in the Sukenobe area and from an underground observatory (Active Fault Survey Tunnel (AFST)) located along the central segment. In the Sukenobe area, fault gouge (samples 1236 and 1236e) and weakly fractured rocks near the fault core (samples 1236a–b and 1105a–b) were collected. In the AFST, two prominent fracture zones along the Mozumi–Sukenobe Fault have been identified[37,41,43]. Fault gouge (sample AFST) was collected from the southern fracture zone B (N68°E, 86°S; Fig. S2). In addition, we collected host rocks (Tetori Group) from areas farther north, including in Sengoku, Kiritani, and Inotani (Fig. 1a). The Sengoku and Kiritani areas are dominated by sandstone, whereas the Inotani area consists of interbedded sandstone and conglomerate with variable grain sizes.

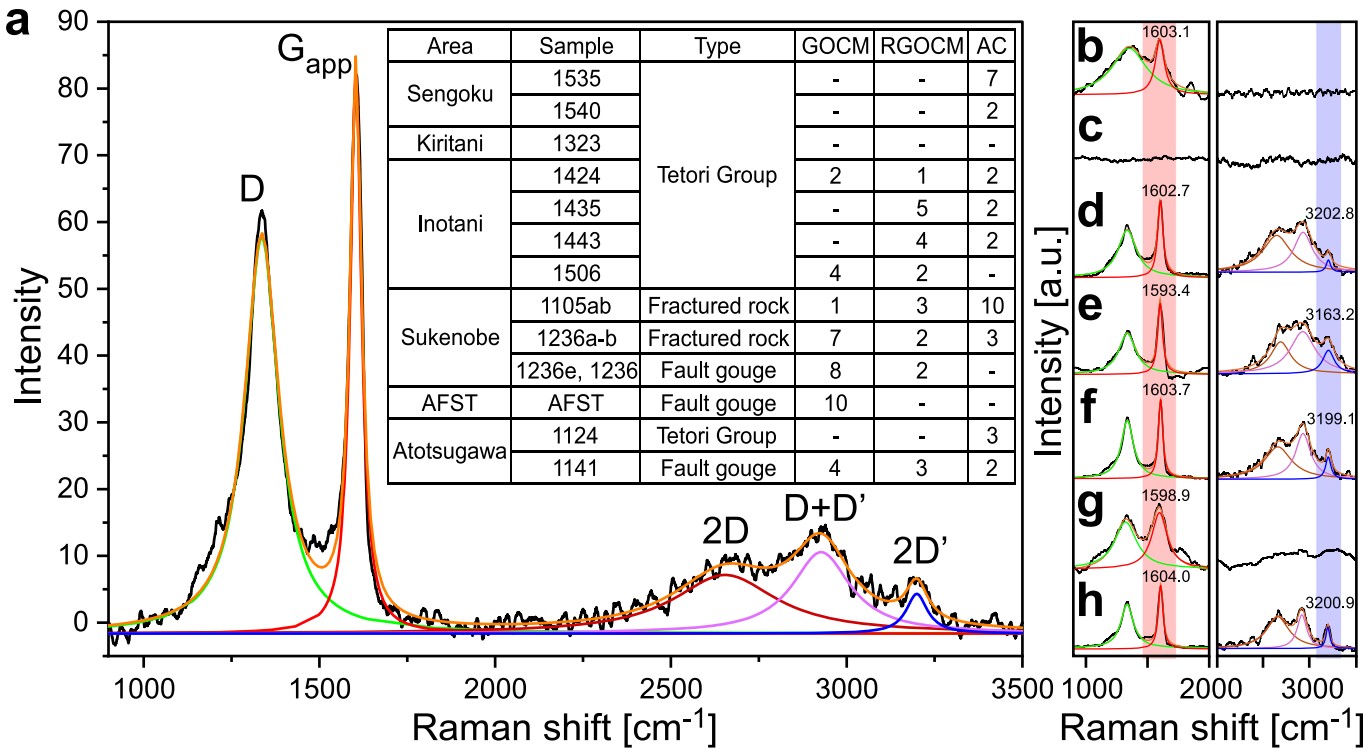

**Fig. 3 | Raman spectra of rocks from the Atotsugawa Fault System. a** Fault gouge from the AFST (sample AFST, spot 5). The apparent G peak ($G_{app}$) and 2D′ peak were used to identify graphene oxide–like CM. The inset table shows the classification of CM in each study area (Tables S2–6). The table shows the total number of analysis spots classified as graphene oxide–like CM (GOCM), reduced graphene oxide–like CM (RGOCM), and amorphous carbon (AC; for which the 2D′ peak could not be not identified). **b** Tetori Group in the Sengoku area (sample 1535, spot 6). **c** Tetori Group in the Kiritani area (sample 1323, spot 1). **d** Tetori Group in the Inotani area (sample 1506, spot 8). **e** Fault gouge in the Sukenobe area (sample 1236, spot 3). **f** Fault gouge in the AFST (sample AFST, spot 5). **g** Tetori Group in the Atotsugawa area (sample 1124, spot 4). **h** Fault gouge in the Atotsugawa area (sample 1141, spot 10).

Notably, bedding planes in the Inotani area (sample 1506) are sub-horizontal. The sample of Tetori Group sandstone from the Inotani area (sample 1506; Fig. 2b) contains quartz, feldspar, and a minor amount of CM. Fault gouge from the Sukenobe area (sample 1236; Fig. 2c) contains only small quantities of very fine- to medium-grained white and gray minerals, including quartz and feldspar. Fragments of black CM grains are dominant and mainly comprise clay- to silt-sized particles. The gouge from the AFST (Fig. 2d) consists of fragments rich in CM, which is uniformly composed of clay-sized particles. XRD analysis of the AFST fault gouge identified quartz, illite, and muscovite based on multiple diffraction peaks (Fig. S9). A peak at $2\theta = 12.4°$ was interpreted to reflect contributions from graphene oxide and chlorite. In three samples from the fault zone (samples 1506, 1236, and AFST), Raman spectroscopy identified graphene oxide–like CM, including graphene oxide and/or its stacks.

Focusing on the AFST gouge, we also investigated the distribution of CM using XPS line analysis, demonstrating that the CM is composed of oxygen-functionalized $sp^2$ carbon and exhibits the same chemical state as graphene oxide. Furthermore, TEM observations confirm the structure of CM along microcracks in the AFST gouge. After drying, the fault gouge sample from the AFST was impregnated with epoxy and cut parallel to the XZ plane based on the strike and dip of the main fault in fracture zone B. The gouge contains a steeply dipping (N18°W, 80°S ~ N32°W, 90°) scaly cleavage defined in thin section by sub-mm, green-black color banding and a preferred orientation of elongated gouge particles (Fig. 2e). SEM observations show networks of microcracks around the edges of the gouge particles and subparallel to the scaly cleavage surfaces (Fig. 2f; blue polygons), while other regions of the gouge are relatively intact (Fig. 2f; dashed blue polygon). The orientation of the cleavage surfaces is consistent with the strike and dip of steeply dipping strata in the Tetori Group around the Atotsugawa Fault System[37,41,43,44].

**Raman spectroscopy**

We measured the Raman spectra of 10 different spots for each randomly selected black CM particle in the samples (Fig. 3a–h). Except for the Kiritani area (Fig. 3c), most of the particles (Fig. 3a, b, d–h) were identified as CM particles based on distinct Raman bands at wavenumbers of ~1350 and ~1600 cm⁻¹. The apparent G peak at ~1600 cm⁻¹ ($G_{app}$) is actually the superposition of two peaks, the G and D′ peaks. When measuring the D′ peak to identify graphene oxide, it is impractical to directly measure the position and intensity of the D′ peak due to its overlap with the G peak. However, the second-order transition (2D′; observed at ~3200 cm⁻¹) remains distinct and is independent of other D bands. By simply halving the energy of the 2D′ mode, we can obtain the energy of the inferred D′ mode. The difference in Raman shifts between the apparent G peak ($G_{app}$) and the inferred D′ peak ($D'_{inf}$) provides information about the presence of graphene oxide[45]. The correlation between $D'_{inf} - G_{app}$ and the redox state (C/O ratio) allows us to apply a quantitative classification of graphene oxide–like CM ("Methods"; ref. 45).

Raman measurements show that all of the studied fault gouge samples contain graphene oxide–like CM (inset in Fig. 3a, e, f, and h). Distinct peaks were identified in 8 out of 10 spectra for the Sukenobe sample, all 10 spectra for the AFST sample, and 4 out of 10 spectra for the Atotsugawa sample. Additionally, a few spectra in each gouge sample show a signal of reduced graphene oxide–like CM, with degrees of oxidation between graphite and graphene oxide (Sukenobe: 2 out of 10; AFST: none; Atotsugawa: 3 out of 10). CM grains without an identifiable 2D′ peak—hereafter referred to as amorphous carbon (AC)—were also identified in the Atotsugawa sample (2 out of

10 spectra). They are more amorphous than graphene oxide–like CM, as indicated by the absence of a 2D′ peak. From the Tetori Group around the Atotsugawa Fault System, CM from the Inotani area contains graphene oxide–like CM with various redox states (inset in Fig. 3a), while CM from other areas contains only AC (inset in Fig. 3a, b, and g).

## X-ray photoelectron spectroscopy

We performed XPS on four samples in which graphene oxide–like CM was identified by Raman spectroscopy (samples 1506, 1236, AFST, and 1141). During XPS, a material is irradiated with weak X-rays and photoelectrons are emitted from atomic orbitals due to the photoelectric effect. The energy of each photoelectron, called the bonding energy, has a characteristic value for each material based on the Fermi level. C 1s and O 1s refer to the XPS spectra corresponding to the 1s orbitals of the carbon and oxygen atoms, respectively. The areas of these peaks indicate the atomic concentrations, and spectral deconvolution allows

analysis of their chemical bonding states. XPS analysis shows that the chemical states of the studied samples consist of $sp^2$ and $sp^3$ carbon bonds, along with oxygen-containing functional groups, including hydroxyl, epoxy, carbonyl, and carboxyl groups (Fig. 4 and Table 1). The $sp^2$ structure describes carbon double bonds and the basal plane of graphene, and the $sp^3$ structure describes carbon single bonds[46,47]. Thus, the structures indicate the predominance of $sp^2$ sheet structures. Furthermore, our analysis indicates that hydroxyl groups are the most abundant (20–40 atm%) oxygen-containing functional groups. In particular, fault gouge from the Atotsugawa Fault exhibits a higher oxygen-to-carbon (O/C) ratio (0.61) than other samples, indicating a higher degree of oxidation in the fault zone. The surface chemical composition and degree of oxidation of the four samples were confirmed to be the same as those of graphene oxide[48,49].

XPS line analysis was used to understand the spatial distribution and concentration of CM within the AFST fault gouge sample (Fig. 5a–c). Line analysis was performed across regions of intact gouge associated with networks of microcracks (Fig. 5a). Figure 5b shows the concentration of carbon and oxygen at each measurement point. The C 1s peaks show a distinctly increased concentration within microcracks (Fig. 5b). Figure 5c shows the concentration of carbon bonds and oxygen-containing functional groups. At spot 4, located within a 100 μm wide crack, the chemical bonding state of carbon is represented primarily by carbon with bound oxygen (Fig. 5b). The presence of epoxy groups in this spot (Fig. 5c) indicates that epoxy resin and oxygen-functionalized $sp^2$ CM are present in approximately equal amounts. At spot 6, microcracks ~5 μm wide contain a relatively high proportion of $sp^2$ bonds and epoxy groups are absent (Fig. 5c). The carbon in these microcracks is identified as CM characterized by $sp^2$ structures bonded with oxygen-containing functional groups, without a contribution from epoxy resin.

## TEM observations

We used TEM to analyze the AFST gouge sample where graphene oxide-like CM was identified by Raman spectroscopy (Fig. 3a) and XPS (Figs. 5 and S10). The thin foil used for TEM was cut from a polished sample containing a microcrack previously analyzed by Raman spectroscopy and XPS prior to FIB fabrication (Fig. 6). In the overview TEM image (Fig. 6a), the dark regions consist mainly of quartz and illite, whereas the brighter regions contain microcracks filled entirely by CM (Fig. S11). The selected area electron diffraction (SAED) pattern from a bright region (inset in Fig. 6a) shows a faint ring with a d-spacing of 0.21 nm, corresponding to the 100 reflection of graphene[50,51]. No other diffraction rings (including 002 of graphite with an interlayer spacing of ~0.34 nm) were observed, indicating that the constituent CM particles are exfoliated graphene oxide with a completely two-dimensional structure (i.e., lacking any stacking). EDS analysis (Fig. 6b) of the area outlined by the dashed blue box reveals a high carbon and oxygen content, consistent with the presence of graphene oxide. High magnification images show a granular structure with particle sizes of <10 nm (Fig. 6c). Individual particles exhibit black-white diffraction contrast depending on their crystallographic orientation (Fig. 6c). HRTEM images of the individual particles reveal several localized regions containing lattice fringes with a d-spacing of 0.21 nm, corresponding to that of (100) of graphene, which is confirmed by the fast Fourier transform images obtained from each region (Fig. 6d). The orientation of the lattice fringes varies between particles and the

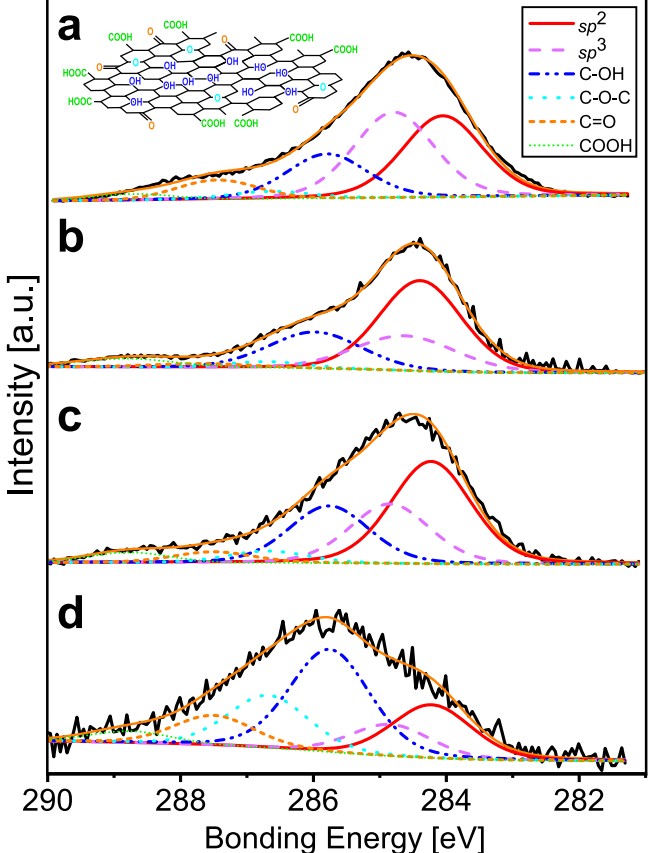

**Fig. 4 | Representative spectra from X-ray photoelectron spectroscopy (XPS).**
**a** Tetori Group from the Inotani area (spot 1506_1). **b** Fault gouge from the Mozumi–Sukenobe Fault in the Sukenobe area (spot 1236_1). **c** Fault gouge from the Mozumi–Sukenobe Fault in the AFST (spot A1_1). **d** Fault gouge from the Atotsugawa Fault in the Atotsugawa area (spot 1141_1). The inset in part A shows the Lerf–Klinowski structural model of graphene oxide[46,47].

**Table 1 | Chemical states determined by X-ray photoelectron spectroscopy (XPS)**

| Area | Rock type | $sp^2$ [atm%] | $sp^3$ [atm%] | C-OH [atm%] | C-O-C [atm%] | C = O [atm%] | COOH [atm%] | O/C ratio |
|---|---|---|---|---|---|---|---|---|
| Inotani | Tetori Group | 31.30 | 29.56 | 23.55 | 4.90 | 8.24 | 2.46 | 0.39 |
| Sukenobe | | 40.85 | 23.86 | 22.52 | 4.08 | 4.52 | 4.16 | 0.37 |
| AFST | Fault gouge | 46.73 | 21.34 | 24.49 | 1.87 | 2.33 | 3.24 | 0.34 |
| Atotsugawa | | 18.94 | 13.77 | 38.18 | 16.47 | 10.52 | 2.13 | 0.61 |

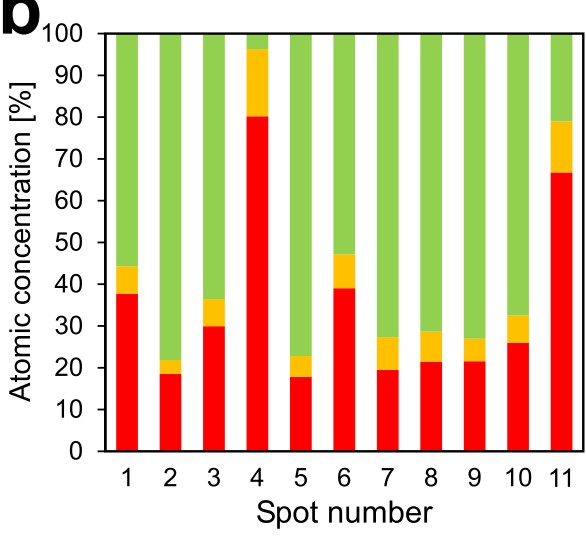

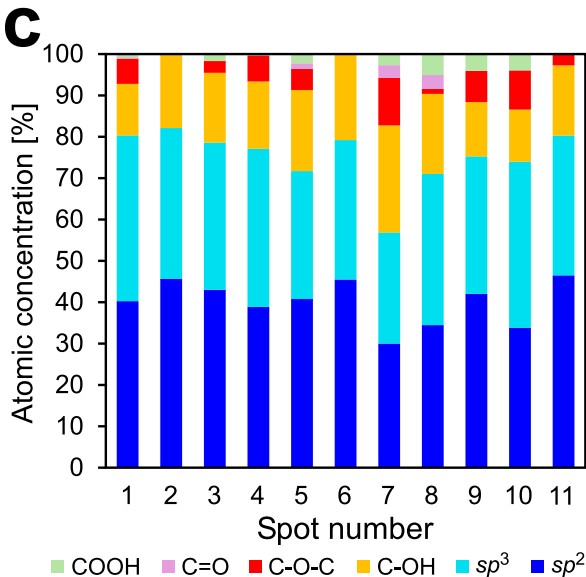

**Fig. 5 | Line analysis by X-ray photoelectron spectroscopy (XPS) across fault gouge and microcracks from region 4 in sample AFST.** See Fig. 2f for location. **a** Scanning X-ray image of region 4. **b** Atomic concentration of carbon and oxygen in the measured points. **c** Concentration of different chemical bonding types in the measured points.

particle size of graphene oxide is estimated to be 3–10 nm from the continuity of the lattice fringes.

On the other hand, the interface between illite and CM has roughness (shown by yellow dashed region in Figs. 6a and S11). A SAED pattern from location 9 is consistent with graphene, whereas that from location 19 shows diffraction patterns attributable to both graphene and clay minerals (Fig. S12). This suggests that graphene oxide and illite may be locally mixed, with some graphene oxide nanosheets potentially arranged along irregular interfaces with illite.

## Discussion

### Identification of graphene oxide

Based on Raman spectroscopy, XPS analysis, and TEM observations, we identified graphene oxide in fault gouge from the Atotsugawa Fault System, one of the most active fault systems in Japan (Figs. 3–6 and Table 1). This represents the first identification of graphene oxide within an active fault. Although CM is widely present in the Tetori Group host rocks, graphene oxide–like CM characterized by graphene oxide and/or its stacks is abundant only in fault gouge from the Atotsugawa Fault System, suggesting that it is concentrated along the fault system. XPS analysis of CM from these samples indicates that their surface chemical composition and degree of oxidation (O/C ratio) are identical to graphene oxide. In addition, the fault gouge from the Atotsugawa Fault is more oxidized than other samples (Fig. 4 and Table 1). This suggests that the Atotsugawa Fault developed under particularly oxidizing conditions or that relatively oxidized graphene was secondarily accumulated within the fault zone. Using XPS line analysis of the AFST fault gouge sample, we demonstrated that CM characterized by $sp^2$ structures bonded with oxygen-containing functional groups is concentrated within microcracks (Figs. 5 and S10). TEM observations from a microcrack confirm that graphene oxide occurs as single-layer sheets with lateral dimensions of 3–10 nm (Fig. 6). Therefore, by combining state-of-the-art Raman spectroscopy, XPS, and TEM, we identified graphene oxide in the fault gouges of the Atotsugawa Fault System.

### Frictional properties and occurrence of graphene oxide

Fault gouge from the AFST contains networks of microcracks around gouge particle boundaries and along scaly cleavage surfaces (Fig. 2e, f). XPS and TEM analyses confirm that the CM concentrated along microcracks is single-layer graphene oxide (Figs. 5 and 6). Although the microcracks may have widened during exhumation or sample preparation, microstructural observations suggest that they preserve the primary gouge fabrics. We therefore propose that the composition of the epoxy-free microcracks is representative of the syn-tectonic cleavage surfaces or gouge particle boundaries. In this case, the graphene oxide nanosheets can enter the frictional interface within the microcracks, preventing direct surface–surface contact between the other constituent phases in the gouge. The friction coefficient of graphene oxide ($\mu \sim 0.01$)[22,23] is at least an order of magnitude lower than that of typical rocks and minerals ($\mu = 0.6–0.85$)[16], relatively low friction minerals such as clays ($\mu = 0.2–0.8$)[16,17], and even graphite ($\mu = 0.1$)[20,21]. Consequently, graphene oxide sheets promote the formation of tribofilms, resulting in extremely efficient lubrication and significant reduction of the friction coefficient[49,52]. We suggest that the presence of graphene oxide along scaly cleavage and microcracks could induce ultra-low friction in fault gouges, potentially leading to the observed aseismic creep and low seismicity along the Atotsugawa Fault System.

The relative abundance of $sp^2$ structures in the graphene oxide (Fig. 5c) suggests either that shear deformation promoted an increase in $sp^2$ or that graphene oxide with a high concentration of $sp^2$ was preferentially concentrated in the microcracks. Notably, previous friction experiments that generated graphene oxide from amorphous carbon demonstrated that tribochemical reactions can promote a

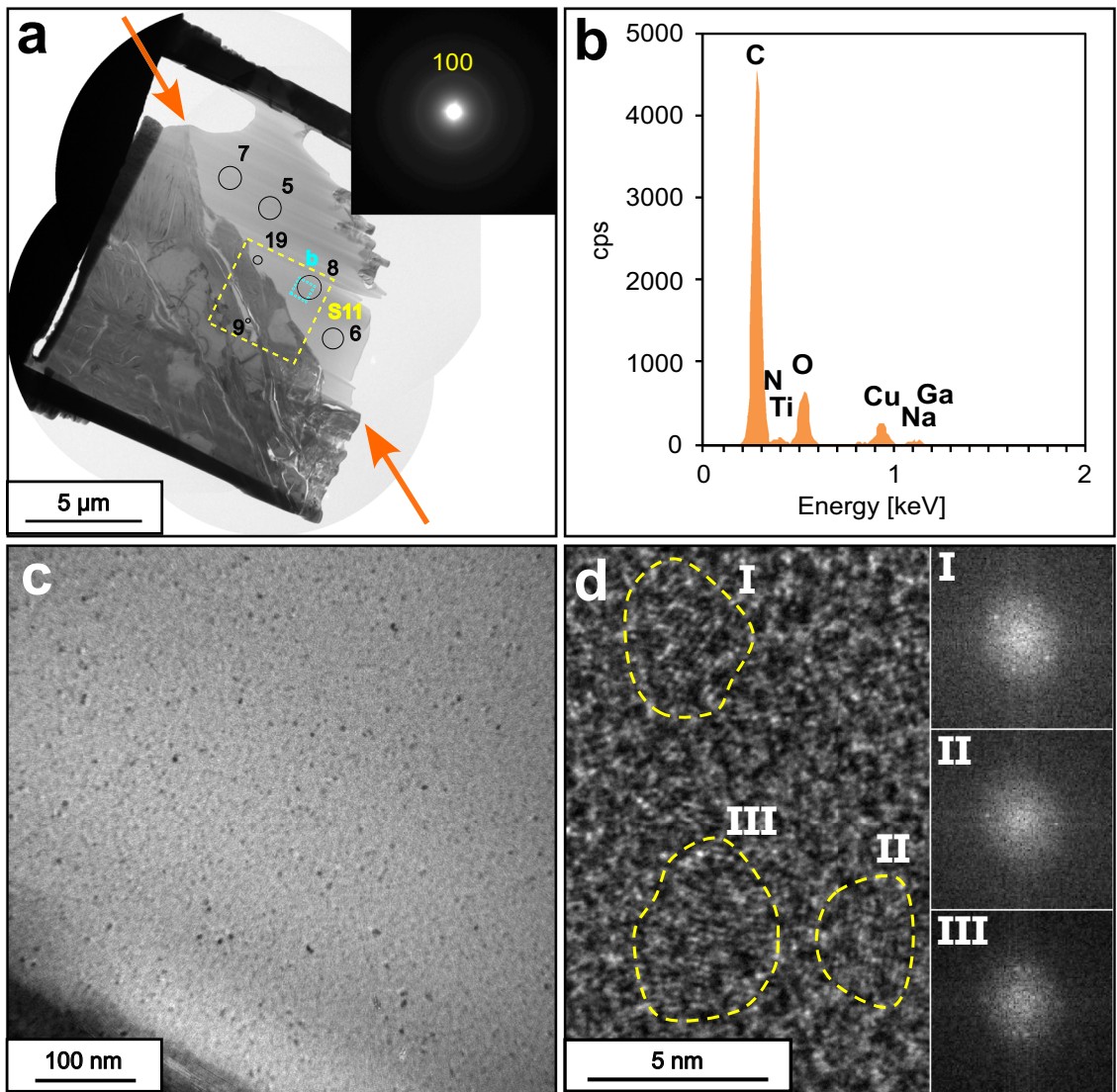

**Fig. 6 | TEM images, diffraction patterns, and EDS analysis of graphene oxide in the AFST fault gouge sample.** The foil sample was cut from a microcrack located in spot 8 of region 9. **a** TEM image of the foil sample. The inset shows a SAED pattern collected from location 7 in the TEM image. The pattern contains a faint ring with a d-spacing of 0.21 nm, corresponding to the 100 reflection of graphene. EDS elemental maps were obtained from the yellow dashed region (Fig. S11). **b** EDS spectrum obtained from the dashed blue region in part (**a**). **c** Detailed TEM image of the microcrack shown in this figure (**a**). The dark speckles show contrast depending on their crystallographic orientation. **d** HRTEM image of the dark speckles. The insets show the fast Fourier transform (FFT) images from regions I, II, and III. The yellow dashed regions contain distinct lattice fringes with a spacing of d = 0.21 nm, corresponding to the 100 of graphene.

change in carbon bonding state from $sp^3$ to $sp^2$. The reactions promoted the spontaneous formation of planar graphene-like structures, resulting in ultra-low friction[22]. We suggest that tribochemical reactions may be important in forming and concentrating graphene oxide with $sp^2$ structures during fault slip.

XPS analysis shows that the chemical state of the graphene oxide is dominated by hydroxyl groups (Fig. 4 and Table 1) bonded to the basal planes of graphene[46,47]. Compared with other oxygen-containing functional groups, hydroxyl groups promote lower friction by forming hydrogen-bond interactions with confined interlayer water[53]. This suggests that graphene oxide enriched in hydroxyl groups provides the most effective chemical state for reducing fault strength.

### Formation and tectonic significance of graphene oxide

Graphene oxide is thermally unstable and loses mass as the temperature approaches 200 °C.[54,55] This is due to the decomposition of oxygen-bearing functional groups, resulting in the generation of gases such as $H_2O$ and $CO_2$. In the seismically active regions, the frictional heating during seismic slip can result in a temperature increase above 200 °C in fault gouges, creating conditions in which graphene oxide becomes unstable. Conversely, graphene oxide may remain stable in low-seismicity or creeping regions, where temperatures remain below 200 °C. This includes the central part of the Atotsugawa Fault System (Fig. 1b). Based on the geothermal gradient around the Atotsugawa Fault System (25–30 °C/km)[56], graphene oxide is predicted to remain stable to depths of approximately 6.3–7.5 km, which corresponds closely to the lower depth limit of the low-seismicity region (Fig. 1b). We suggest that graphene oxide is preferentially preserved in low-seismicity or creeping regions along the Atotsugawa Fault System, and that this provides a mechanistic link between carbon chemistry, shear localization and fault slip behavior.

EDM (1981–1999) and GPS (1997–2001) observations revealed fault creep along the central part of the Atotsugawa Fault System[1,2,11,12]. However, later GPS measurements indicate that the same fault segment was locked from 1998 to 2006, revealing an apparent inconsistency between the datasets[2]. InSAR and GNSS analyses further demonstrated an abrupt

increase in velocity along the central segment[13]. In addition, the interseismic strain-rate gradient varies over time, and cumulative displacements do not increase proportionally[13]. These data have been interpreted to reflect temporal changes in seismicity along the central part of the fault, possibly associated with local stress variations in the seismogenic zone at both fault tips[14]. Taken together, these observations suggest that time-dependent variations in fault slip behavior should be considered to explain the distinctive slip history of the Atotsugawa Fault System.

Shear during fault slip causes an evolution in gouge microstructure, including the disruption and rearrangement of particles along and between scaly cleavage surfaces. These processes could promote the formation and redistribution of graphene oxide during slip, implying that the lubricating effect of graphene oxide likely changes over time. This temporal variability in microstructure and frictional behavior suggests that the formation and evolution of graphene oxide may influence time-dependent fault creep. However, testing this hypothesis requires further geodetic and seismological observations combined with experimental constraints on the frictional evolution of graphene oxide–bearing fault gouges.

Seismological observations reveal a low-velocity anomaly along the central segment of the Atotsugawa Fault System (Fig. 1b). This anomaly has been linked to the circulation of deep fluids along the fault and has been proposed as an explanation for fault creep[9]. In this region, the abundance of water in cracks and fractures at relatively low temperatures is expected to form graphene oxide. Due to the interaction between water and CM, the formation of interlayer water in graphene oxide promotes the formation of hydroxyl groups during reactions at temperatures below 200 °C.[57] The observed chemical state of graphene oxide in the Atotsugawa Fault System, characterized by a high concentration of hydroxyl groups, can therefore be explained by three interacting factors: temperatures below 200 °C, tribochemical reactions induced by fault movement, and the presence of interlayer water.

Previous geological studies of graphene and graphite suggest that two main processes may be responsible for the formation of graphene oxide[32–34]. (1) Amorphous carbon derived from organic matter transforms into graphene oxide through graphitization accompanied by oxidation during fault slip. (2) Crystalline graphite undergoes exfoliation and oxidation during shearing to produce graphene oxide. These processes include CM of organic origin, together with inorganic chemical reactions[58]. In the Atotsugawa Fault System, both processes could be responsible for the formation of graphene. This is because the widely distributed sediments of the Tetori Group contain organic carbon, and there is an abundant fluid supply from depth (Fig. 1; ref. 9).

We propose that graphene oxide is more likely to have formed through the transformation of amorphous carbon derived from organic matter during fault-induced tribochemical reactions. Despite extensive efforts to exfoliate crystalline graphene in materials science experiments[54,59], complete exfoliation remains challenging. Some degree of layering likely persists during the exfoliation process, consistent with observations of graphene in geological settings[32–34]. However, our TEM observations (Fig. 6) show that graphene oxide in the Atotsugawa Fault System occurs as individual sheets with lateral dimensions of 3–10 nm. This implies that graphene oxide formed directly from amorphous carbon and that tribochemical processes can promote enlargement of graphene oxide sheets. This interpretation is consistent with models of graphite enrichment along the Atotsugawa Fault System[15]. The fault gouge contains clay minerals such as illite (Figs. S9, S11, and 12). Previous studies in materials science have demonstrated that platy minerals interact with graphene and contribute to its formation[60,61]. The formation of graphene oxide in faults may reflect a bottom-up approach involving tribochemical reactions and mineral surface interactions, rather than top-down approach such as exfoliation of graphite. This remains an important subject for future investigation.

Carbon-bearing fault rocks have been found within intraplate faults[15,58], along subduction zone plate-boundary faults, and within adjacent seafloor sediments[62–65]. Diagenetic reactions during subduction increase the crystallinity of CM and weaken faults with depth[62]. For example, graphite-bearing slip zones have been identified in a fossil subduction megathrust within the Shimanto accretionary complex in Japan[63]. Friction experiments have demonstrated that shearing amorphous carbon can trigger tribochemical reactions that form ultra-low friction graphene oxide films[22]. Our work documents graphene oxide in a natural fault zone for the first time. This suggests that shear-induced transformations in fault zones can lead to the formation of graphene oxide, dramatically reducing the frictional strength of the fault and potentially leading to aseismic slip.

## Methods

We collected samples from the Atotsugawa Fault System and used Raman spectroscopy to search for graphene oxide–like CM (GOCM; Fig. 3 and Tables S2–S6). We used XPS to examine the chemical states of the CM, including the types and quantities of oxygen-bearing functional groups (such as hydroxyl and carboxyl groups), as well as the overall oxygen content (Fig. 4 and Table 1). This approach enables identification of carbon consisting of $sp^2$ structures modified by oxygen-containing functional groups. To investigate the distribution and structure of graphene oxide in the fault gouges, we conducted XPS line analysis (Figs. 5 and S10), SEM (Fig. 2f), and TEM observations (Fig. 6). By combining Raman spectroscopy with XPS and TEM, we were able to identify single-layer graphene oxide with a high degree of confidence.

### Raman spectroscopy

For Raman spectroscopy, samples were cut to ~1 cm in thickness and several centimeters in width and length. They were dried and the surfaces were polished by hand using waterproof abrasive paper (#180, #1000, and #2000). Raman spectra were acquired using a JASCO NRS-5100AMS Raman spectrometer in the Technical Division, School of Engineering, Tohoku University, Sendai, Japan. Analyses were conducted using 532.3 nm laser excitation, with the intensity adjusted to 0.5 mW at 5% power to avoid sample damage. The low laser power is unlikely to induce defects associated with the D′ band, as reported for higher power settings[66,67]. The measurements were conducted by acquiring spectra over a frequency range of 900–3500 cm⁻¹, with two scans taken with exposure times of 10 s each. The wavenumber resolution was 0.4 cm⁻¹. To avoid polishing-induced alterations in the Raman spectra, especially shifts in the apparent G band position[35], polishing was performed manually using relatively coarse sandpaper (~#2000) and measurements involved focusing the laser beam a few micrometers below the surface. OriginPro was used to process spectra by performing Savitzky–Golays smoothing operations. Baseline correction was applied using a third-order polynomial for the primary Raman modes in the 900–2000 cm⁻¹ range, and a linear function for the secondary Raman modes in the 2000–3500 cm⁻¹ range. For spectral analysis, each peak was fitted using a Lorentzian function.

When measuring the Raman shift of the D′ peak for the identification of graphene oxide–like CM (GOCM), it is impractical to directly measure the position and intensity of the peak due to its overlap with the Raman shift of the G peak. However, the second-order transition (2D′; observed at ~3200 cm⁻¹) remains distinct and independent from other D bands. By halving the energy of the 2D′ mode, we can obtain the energy of the inferred D′ mode. Therefore, the difference between the Raman shifts of the apparent G peak at ~1600 cm⁻¹ ($G_{app}$) and the inferred D′ peak ($D'_{inf}$) based on the Raman shift of the 2D′ peak (overtone of the D′ peak) provide information about the presence of graphene oxide–based nanoplatelets (GONP)[45]. On this basis, we quantitatively classified the graphite materials into graphene oxide–like CM (GOCM, $D'_{inf} – G_{app} < 0$ cm⁻¹), reduced graphene oxide–like CM (RGOCM, $0 < D'_{inf} – G_{app} < 25$ cm⁻¹), and graphite ($D'_{inf} – G_{app} > 25$ cm⁻¹). According to Raman spectroscopy analysis[45], increasing the density of

defects through oxidation causes the D′ peak to shift to a lower energy than the G peak, and therefore D′$_{inf}$ – G$_{app}$ is decreased by oxidation.

It has been reported that the 2D-band peak shifts upwards in the Raman spectra as the number of graphene layers increases[68]. However, the 2D-band peak appears in monolayer graphene as well, so variations in this peak can only be used as a relative indicator. Natural samples of graphene and graphene oxide may be wrinkled and lack well-ordered orientations. Consequently, when estimating the number of graphene layers in natural samples, the 2D-band peak is expected to show significant variability[69]. For this reason, Raman spectroscopy cannot accurately determine the thickness of graphene layers. Nevertheless, measurements of graphene oxide produced by various methods[47] demonstrate that Raman spectroscopy is a simple and reliable approach for identifying graphene oxide. Strictly speaking, graphene is a single-layer structure by definition. Considering the possibility that the analyzed CM may consist of one or more layers, the terms "graphene oxide–based" or "graphene oxide–like" can be used. In this paper, we refer to the studied as graphene oxide–like CM (GOCM) and reduced graphene oxide–like CM (RGOCM).

### X-ray photoelectron spectroscopy (XPS)

For XPS, fragments several millimeters in size were extracted from the samples, and the fragment surfaces were polished. These samples were analyzed to identify the chemical states of the CM. Quantitative XPS analysis was conducted using a PHI-5000 VersaProbe II (Ulvac PHI) at MaSC, Tohoku University, Sendai, Japan. Narrow scans were performed at the highest C1$s$ peak intensities (from 3 to 5 measurement points) and the average chemical states were determined. The X-ray source for XPS was AlKα (1486.6 eV). The pass energy for qualitative analysis was set to 117.4 eV, with a step size of 1 eV. For narrow scans (chemical state analysis), the pass energy was set to 58.7 eV, with a step size of 0.1 eV over the range of 278–291 eV. For quantitative XPS analysis, the spot size was 100 μm. The C1$s$ peaks in the XPS spectra were deconvoluted into $sp^2$, $sp^3$, hydroxyl groups (C–OH), epoxy groups (C–O–C), carbonyl groups (C = O), and carboxyl groups (COOH) using the instrument-bundled software or OriginPro. The O/C ratios were also calculated[30,31]. XPS analysis was performed using the software bundled with the XPS instrument or OriginPro.

For XPS line analysis, the fault gouge sample from the AFST was used to investigate the distribution of CM characterized by $sp^2$ structures bonded with oxygen-containing functional groups. The sample was impregnated with epoxy and cut along the XZ plane based on the strike and dip of the main fault in fracture zone B (the most fractured part III, N68°E, 86°S; Fig. S2). The cut surface was polished by hand using waterproof abrasive papers (#180, #1000, #2000, -0.5 μm). XPS line analysis was conducted across microcracks in nine distinct regions, with Scanning X-ray images used to accurately determine the positions of the analyses. During line analysis, the spot size for region 1 was 100 μm, for regions 2–8 it was 20 μm, and for region 9 it was 10 μm. All other measurement conditions were identical to the quantitative XPS analysis.

The natural CM in the samples was distinguished from the CM in the epoxy resin by the absence of epoxy groups. In the resin used in this study, the atomic ratio of epoxy groups to hydroxyl groups is 4:3. If carbon is sufficiently bonded to oxygen-containing functional groups and no epoxy groups are present, the analyzed material can be classified as graphene oxide only. When the epoxy group content is significantly lower than the hydroxyl group content, the analyzed material contains a relatively small amount of epoxy resin and a relatively large amount of CM characterized by $sp^2$ structures bonded with oxygen-containing functional groups.

### SEM and TEM observations

Samples for SEM and EDS analysis were coated with a 5-nm-thick layer of Os. SEM imaging and energy dispersive X-ray spectroscopy (EDS) were carried out with a field-emission scanning electron microscope (FE-SEM, JEOL JSM-7001F) in the Department of Earth Science, Tohoku University, Sendai, Japan. The analyses were carried out using an accelerating voltage of 15 kV and a working distance of 10 mm.

To investigate the characteristics of the graphene oxide (e.g., size and structure), TEM was performed using a JEOL JEM-2100F in the Geodynamics Research Center, Ehime University, Matsuyama, Japan. A thin foil was prepared using a focused ion beam system (Thermo Fisher Scientific Scios 2) at Ehime University. The foil was extracted across a microcrack from spot 8 in region 9, which represents a higher-magnification view of spot 6 of region 4 in the AFST gouge sample, which had previously been examined using XPS line analysis. TEM observations, including high-resolution TEM (HRTEM) imaging, SAED, and EDS analysis in scanning TEM mode, were conducted at an accelerating voltage of 200 kV.

### X-ray diffraction (XRD) analysis

For XRD analysis, the AFST fault gouge was powdered using a planetary ball mill. XRD analysis was performed using an automated multi-purpose X-ray diffractometer (Philips, X'Pert-PRO) at the Department of Earth Science, Tohoku University, Sendai, Japan, under conditions of 40 kV and 50 mA. Mineral identification was based on XRD peaks with reference to Figs. 6, S11, and S12, and ref. 39.

## Data availability

All analytical results obtained are provided in the main text and the Supplementary Information. The raw data include results derived from rock samples collected on land and within tunnels owned by a mining company, and access to these data may be subject to restrictions related to ownership. As noted in the Acknowledgements, permission was obtained from the relevant company. The data can be made available on request, subject to these restrictions.

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

## Acknowledgements

This work was supported by KAKENHI grant numbers 22H04932, 26K21718 (J.M.), 24K00724 (H.N.), and 25KJ0633 (T.S.). We thank the Tohoku University APC Support Program for its support in covering the Article Processing Charge. We acknowledge A. Suzuki, T. Takanashi, and M. Nemoto for their assistance with the Raman measurements and Y. Gambe for his invaluable support with the XPS analysis. We thank M. Tanaka and H. Nishihara for constructive comments and suggestions, which improved the quality of this study. We also thank H. Takagi for providing images of the Atotsugawa Fault System. We are grateful to N. Sakurai (President and CEO of *Kaiyo Shuppan* Co., Ltd.) and T. Ito for granting permission to reproduce the outcrop sketch in the Active Fault Survey Tunnel (AFST). We also sincerely thank I. Cho of National Institute of Advanced Industrial Science and Technology (AIST) for granting permission to reproduce figures from the Active Fault Database of Japan. P. He (Power Reactor and Nuclear Fuel Development Company), K. Otsuki, and H. Ito helped N. Nakamura with the field survey of the active fault in the underground tunnel. R. Maekawa (Director and General Manager of the Mining Division, Kamioka Mining and Smelting Company Limited) granted us permission to publish the details and analytical results of samples collected from the active fault survey tunnel. Additionally, we are extremely grateful to T. Takenawa and K. Ogita for their assistance with sampling. J.M. discloses support for the research of this work from Funder [22H04932, 26K21718]. H.N. discloses support for the research of this work from Funder [24K00724]. T.S. discloses support for the research of this work from Funder [25KJ0633]. This work was supported by publication funding from the Tohoku University APC Support Program.

## Author contributions

T.S., H.N., and J.M. designed and conceived the study. T.S., J.M., N.N., and H.O. prepared the samples. T.S., J.M., S.S., and H.O. performed Raman spectroscopy, XPS, SEM, and TEM. All authors participated in data and image analysis and contributed to the discussion of the results.

## Competing interests

The authors declare no competing interests.
