## [Transparent Peer Review file · Nature Communications]

Ultra-low friction graphene oxide in the Atotsugawa Fault System

Corresponding Author: Mr Tomoya Shimada

Version 0:

Reviewer comments:

Reviewer #1

(Remarks to the Author)
Reviewer comments on

“Ultra-low friction graphene oxide in the Atotsugawa Fault System”

Submitted to Nature Communications.

The authors integrate detailed field observations from the Atotsugawa Fault System, Japan, with advanced Raman spectroscopy and complementary analytical methods to characterize carbon-bearing materials within both undeformed host rocks and fault zones. The combination of state-of-the-art Raman spectroscopy with additional analyses (e.g., X-ray photoelectron spectroscopy, XPS) allows the authors to identify, for the first time, the presence of graphene oxide in an active fault system.

While the occurrence of low-friction graphite and amorphous carbon in fault zones has been previously reported, the discovery of graphene oxide is particularly significant because of its ultra-low frictional properties. This finding has important implications for understanding shear localization and slip behavior (seismic vs. aseismic slip) in active fault systems. The authors further discuss how these observations may explain the aseismic creep and the seismic gap (up to 6–7 km depth) observed along the Atotsugawa Fault System.

Overall, this study presents potentially important findings that merit publication in Nature Communications. However, I have several concerns regarding the presentation and structure of the manuscript, particularly in the way the results and their geological and seismological context are developed. A clearer, more systematic organization of the results and a more detailed discussion of the seismo-tectonic framework would substantially improve readability and strengthen the manuscript's scientific impact.

Main Issues

1) Presentation of Results

The main objective of the fieldwork was to collect samples from both the protolith (host rocks of the Atotsugawa Fault System) and fault zones, in order to characterize the nature of carbon materials (e.g., amorphous carbon, graphite, graphene oxide) using advanced Raman spectroscopy.

Currently, the presentation of field observations (starting around line 70) is difficult to follow. The text reads as a continuous narrative that jumps between faults and localities without a clear structure, making it hard for the reader to relate the descriptions to Figure 1a. I recommend restructuring this section to present each fault and sampling site systematically, for example:

- Describe each fault separately, identifying both undeformed (protolith) and deformed (fault gouge) samples.
- Summarize the field observations and key analytical findings for each location before moving to the next.

This reorganization would greatly improve the logical flow, clarity, and accessibility of the results.

2) Seismo-tectonic Setting: Provide More Detail and Integration

The introduction effectively outlines the overall research motivation, linking the presence of ultra-low friction graphene oxide to the aseismic behavior of the Atotsugawa Fault System. However, more detail about the seismo-tectonic background is needed both in the Introduction and later in the Discussion to fully contextualize the results.

Specifically:

- Please clarify what is meant by “heterogeneous creep (aseismic slip)” (line 21). Does this imply that most segments of the Atotsugawa Fault System creep, but that some still produce occasional moderate to large earthquakes?
- The seismicity record (Fig. 1b, covering 1998–2010) shows a clear seismic gap, but the manuscript also refers to an historical earthquake in 1858. Providing the magnitude and additional context for this event would be valuable.
- I suggest including information from recent geodetic and seismological studies that quantify fault coupling from, for example, GPS-derived slip rates and spatial variations in creep behavior along the fault system. Adding a new figure summarizing these data (e.g., updated Fig. 1) would help readers visualize the structural and kinematic framework.

In the Discussion, consider exploring whether variations in graphene oxide abundance correlate with differences in slip behavior between fault segments (i.e., creeping vs. seismogenic zones). Since graphene oxide is thermally unstable above ~200 °C, it might be preferentially preserved in (shallow?), creeping segments and absent in (deeper?), seismically active ones, as frictional heating during seismic slip will produce temperature rise well above 200 degrees in the gouges. This relationship could provide an important mechanistic link between carbon chemistry, shear localization, and fault slip behaviour.

To summarise, this is a novel and promising contribution that identifies graphene oxide as a potential control on fault slip behavior. To maximize its impact, the manuscript would benefit from:

- A clearer, more structured presentation of field and analytical results;
- A more comprehensive and better-integrated discussion of the seismo-tectonic setting; and
- A concise discussion about any potential/inferred link between graphene oxide formation, its thermal/mechanical stability, and the observed creeping vs. seismic behavior of the fault system.

Reviewer #2

(Remarks to the Author)

The authors present an interesting study on aspects of nanolith geology, specifically the geology of graphene, with implications for the frictional behavior of shear zones located in Japan. The work has merit in its focus on the identification of natural graphene. However, the material identified does not adequately meet the IUPAC standards and formal definitions required to be classified as graphene oxide or reduced graphene oxide, including the absence of a characterization technique capable of determining the material's thickness. Consequently, the authors' main finding is not fully supported by the presented data—though this does not diminish the scientific value of the study.

The frictional considerations and interpretations regarding the behavior of the graphite nanoplatelets (graphene oxide-like or -based) are indeed relevant and should become the main focus of the revised manuscript, including the presentation and assessment of the primary coefficient of friction data identified by the authors, which may lead to meaningful interpretations of fault movement processes with significance for the scientific and academic community.

Further specific and complementary comments on the manuscript are provided below:

When the authors state “When mixed with quartz, a major mineral in the crust,” it would be advisable to specify that it refers to the continental crust.

The authors should be cautious with the use of expressions such as “single layers of graphene,” since, strictly speaking, graphene is intrinsically a single-layer structure throughout its definition. Any material composed of more than one layer is, by definition, a graphite nanoplatelet, which may exhibit behaviors analogous to those of graphene or graphene oxide, but should not formally be referred to by these terms.

Although the caption of Figure 1 explains what is being represented, this does not eliminate the need to include a legend within the figure itself, for two main reasons: (i) some readers may be color-blind or have other conditions that prevent them from accurately distinguishing colors; and (ii) every figure must be entirely self-contained so that a reader can interpret it without any external reference, which requires the inclusion of a caption within the figure. In addition, the meaning of the area outlined with a blue dashed line between X–X' was not described, not even in the caption.

I did not understand what the authors meant in line 79: "In this paragraph, we summarize geochronological data from each sampling area," since no geochronological data or crustal evolution based on isotopic data are actually presented. Instead, the paragraph describes the location contexts of the sites where the samples used in the study were collected.

In line 95, what does the "black material" refer to? Does it also correspond to a CM? It is important to provide a clearer description of this material, which is currently much less well characterized than the other samples.

The same issue regarding the use of colors and the absence of an internal legend in Figure 1 also occurs in Figure 2. Therefore, the authors will need to revise the way this figure is presented.

Personally, I would prefer the Methods section to be placed immediately after the Introduction in the manuscript, since there are several moments in the text—particularly in the "Results" section—where the authors describe sampling and treatment procedures that are only explained at the end of the paper, which disrupts the textual cohesion. Although the authors clearly made an effort to present the main methodological aspects and to transparently provide the remaining details as supplementary materials, I found the absence of a table compiling all collected samples and their correlation with the evaluated shear zone noticeable. Such a table could be included as part of the supplementary materials.

The oxidized materials identified by the authors do not fully meet the most rigorous and conservative standards currently accepted for classifying a substance as graphene oxide or reduced graphene oxide. The authors should instead refer to the characterized and diagnosed material as oxidized graphite nanoplatelets and reduced oxidized graphite nanoplatelets. At most, to emphasize the finding, the terms graphene oxide-based or graphene oxide-like could be used. Both the Raman spectroscopy results (which show the persistence of a significant 2D band) and the XPS data (indicating a substantial amount of carbon atoms maintaining sp^3 hybridization) suggest that these structures are not definitively monolayered. Therefore, they cannot presently be accurately referred to as graphene—or as its oxide or reduced oxide forms.

At present, all geological substances of this type are considered graphene-like, since the intrinsically complex nature of metamorphic environments would make it highly unlikely for structures to be preserved that fully satisfy the chemical criteria and formalism required to be classified as "true" graphene or graphene oxide. Therefore, the main finding of the authors' work does not actually reflect the extraordinary nature initially suggested. If the authors had indeed identified a natural substance that fully meets the criteria for classification as graphene oxide, the discovery would be extremely significant and of high impact—but this is not supported by the data presented in the study.

A recommendation to the authors is to conduct a thorough literature review focused on the geology of graphene, in order to accurately present the current state of the art in their study. Although the authors employ analytical techniques that are pertinent and appropriate for the work, these methods alone do not reflect advances in geological science regarding the identification of geological graphene-like structures, particularly those found in metamorphic environments. Some examples of relevant studies that the authors should consult and use to update their literature review include:

https://doi.org/10.1007/978-3-031-04435-9_34

<https://doi.org/10.1038/s41561-025-01803-3>

<https://doi.org/10.1007/s12040-025-02619-w>

<https://doi.org/10.1002/chem.201500116>

<https://doi.org/10.1590/1517-7076-RMAT-2022-0122>

<https://www.taylorfrancis.com/chapters/edit/10.1201/b19679-12/natural-graphene-based-shungite-nanocarbon-natalia-rozhkova-sergey-rozhkov-andrey-goryunov>

<https://doi.org/10.1134/S1028334X20110112>

<https://doi.org/10.1002/jrs.6474>

<https://doi.org/10.3390/c9030075>

<https://doi.org/10.1080/19475411.2014.885913>

The frictional properties proposed for the graphene oxide-like or -based material identified by the authors appear to be associated with quantum phenomena that are not observable at the bulk scale in graphite. This is a relevant finding and could be further explored as the main focus of the study.

To support the estimation of the thickness of the analyzed nanoplatelets, it would be advisable to perform a transmission electron microscopy (TEM) or atomic force microscopy (AFM) analysis, which would allow for a more accurate determination of the thickness of the discovered material.

One question that does not seem to have been sufficiently discussed or clarified by the authors—and which is relevant to the overall discussion—is whether the amorphous carbonaceous precursor material crystallized directly to form the graphene oxide-like or -based nanoplatelets exhibiting behavior analogous to graphene oxide, or whether a thicker (bulk) graphitic material first crystallized and was subsequently cleaved and oxidized during shear deformation. This distinction may have important implications for structural geology and for the characterization of the properties of nanoliths.

Version 1:

Reviewer comments:

Reviewer #1

(Remarks to the Author)

Dear Editor,

I have now gone through the revised version of the manuscript and figures, and I find that the paper has improved significantly from the previous version I looked at.

The structure of the manuscript, the logical presentation and discussion of the arguments and the clarity of the data and observations presented in the figures are of an appropriate standard for a journal like Nature Communications.

The Authors have thoroughly addressed my previous comments and criticisms, and took action to improve the text and figures accordingly.

I do not have any further comments or suggestions on the paper and its scientific merit at this stage of the peer review process.

Regards,
Nicola De Paola

Reviewer #2

(Remarks to the Author)

The authors demonstrated a strong commitment to addressing each point and question raised during the first round of review, which is commendable. The manuscript has improved in quality as a result of these efforts. Additional and complementary comments are presented below:

In line 42, the expression “a major mineral in the continental crust” is unnecessary. It would be far more relevant to the discussion to clearly present the principal and accessory mineralogy, and ideally the mineral paragenesis. Is the rock composed exclusively of graphite and quartz? If so, the absence of phyllosilicates or other platy minerals would make the systematic cleavage behavior of the material even more intriguing and worthy of deeper discussion. The presence of platy minerals is well known to play a fundamental role in processes aimed at exfoliating graphite into graphene-like structures. This aspect requires further elaboration, particularly in the context of top-down approaches for the production of graphene-like materials. It is essential to clarify whether the fundamental process responsible for the formation of these crystalline structures should be interpreted as top-down or bottom-up, as this distinction has significant implications for the scientific interpretation and technological relevance of the study.

The authors report the presence of illite in the regions where nano-scale structures analogous to graphene oxide were identified. It is essential to characterize the micro- to nano-structural and textural relationships between the carbonaceous material and the clay mineral, as this association may be critical for constraining the genetic interpretation of the processes responsible for the formation of these crystalline structures. A more detailed analysis of these relationships would substantially strengthen the discussion and enhance the potential scientific impact of the manuscript within the broader research community.

Overall, the revisions implemented by the authors are relevant and contribute positively to improving the discussion. The remaining task is to address the points raised above in order to better focus and centralize the genetic discussion of the nano-structures.

Tracking number: NCOMMS-25-45653-T

Dear Editor and reviewers,

We thank the reviewers for their comments, which have improved the overall clarity and quality of our work. Below, we provide our point-by-point responses. In this document and in the revised manuscript, our replies to Reviewer #1 are shown in blue and our replies to Reviewer #2 are shown in orange.

Should you have any further comments or questions, we would be pleased to address them.

Reviewer #1 (Remarks to the Author):

Reviewer comments on

“Ultra-low friction graphene oxide in the Atotsugawa Fault System”

Submitted to Nature Communications.

The authors integrate detailed field observations from the Atotsugawa Fault System, Japan, with advanced Raman spectroscopy and complementary analytical methods to characterize carbon-bearing materials within both undeformed host rocks and fault zones. The combination of state-of-the-art Raman spectroscopy with additional analyses (e.g., X-ray photoelectron spectroscopy, XPS) allows the authors to identify, for the first time, the presence of graphene oxide in an active fault system.

While the occurrence of low-friction graphite and amorphous carbon in fault zones has been previously reported, the discovery of graphene oxide is particularly significant because of its ultra-low frictional properties. This finding has important implications for understanding shear localization and slip behavior (seismic vs. aseismic slip) in active fault systems. The authors further discuss how these observations may explain the aseismic creep and the seismic gap (up to 6–7 km depth) observed along the Atotsugawa Fault System.

Overall, this study presents potentially important findings that merit publication in Nature

Communications. However, I have several concerns regarding the presentation and structure of the manuscript, particularly in the way the results and their geological and seismological context are developed. A clearer, more systematic organization of the results and a more detailed discussion of the seismo-tectonic framework would substantially improve readability and strengthen the manuscript's scientific impact.

Reply) Thank you very much for your comments. In the following paragraphs, we address each of these comments individually. In particular, the revised manuscript contains a new and expanded discussion of the region's seismo-tectonic framework, as well as the potential role of graphene oxide in controlling fault slip behavior.

Main Issues

1) Presentation of Results

The main objective of the fieldwork was to collect samples from both the protolith (host rocks of the Atotsugawa Fault System) and fault zones, in order to characterize the nature of carbon materials (e.g., amorphous carbon, graphite, graphene oxide) using advanced Raman spectroscopy.

Currently, the presentation of field observations (starting around line 70) is difficult to follow. The text reads as a continuous narrative that jumps between faults and localities without a clear structure, making it hard for the reader to relate the descriptions to Figure 1a. I recommend restructuring this section to present each fault and sampling site systematically, for example:

- Describe each fault separately, identifying both undeformed (protolith) and deformed (fault gouge) samples.
- Summarize the field observations and key analytical findings for each location before moving to the next.

This reorganization would greatly improve the logical flow, clarity, and accessibility of the results.

Reply) We thank the reviewer for this comment. We agree that a more systematic description of each fault and sampling locality is essential, and we have revised the manuscript accordingly. Specifically, in lines 101–123 (and following this), we have reorganized the results to describe the sampling sites grouped by fault rather than by geographic area. We have also revised this section to summarize the key fieldwork results while presenting the detailed descriptions and interpretations later in separate sections focusing on Raman spectroscopy, XPS and TEM analysis. Additionally, we have expanded the supplementary information, which now contains a structured and systematic description of each

sampling locality (Fig. S1).

2) Seismo-tectonic Setting: Provide More Detail and Integration

The introduction effectively outlines the overall research motivation, linking the presence of ultra-low friction graphene oxide to the aseismic behavior of the Atotsugawa Fault System. However, more detail about the seismo-tectonic background is needed both in the Introduction and later in the Discussion to fully contextualize the results.

Reply) We thank the reviewer for this helpful comment. We have expanded our descriptions of the aseismic domain, the overall seismo-tectonic context, and the potential creeping behavior of the Atotsugawa Fault System in both the introduction and the discussion. Specifically, the first paragraph of the introduction now includes a more detailed synthesis of geodetic and seismological observations characterizing the low-seismicity region (lines 21–25) and the inferred creep behavior of the fault (lines 25–33). Building on this expanded introduction, we incorporated additional seismo-tectonic information into the discussion (lines 288–290 and 311–312). We have also added relevant text to the section on the formation and tectonic significance of graphene oxide (lines 318–328 and 337–339).

These revisions more clearly describe the temporally variable creep behavior based on GPS and EDM analyses, seismic velocity tomography (Nakajima et al., 2010), InSAR observations (Takada et al., 2018), and earthquake focal mechanism analyses (Katsumura et al., 2010). By more carefully integrating these seismo-tectonic datasets into our revised results and discussion sections, we have better contextualized our findings and clarified the possible links between graphene oxide and the slip behavior of the Atotsugawa Fault System. Our revised conclusions provide a broader perspective on the potential impact of graphene oxide in fault zones and will therefore appeal to a wider readership.

Specifically:

- Please clarify what is meant by “heterogeneous creep (aseismic slip)” (line 21). Does this imply that most segments of the Atotsugawa Fault System creep, but that some still produce occasional moderate to large earthquakes?

Reply) We thank the reviewer for this comment and agree that our use of the term was ambiguous. We intended to describe the pattern of present-day seismicity, which is characterized by a low-seismicity region in the central fault segment, flanked to the east and west by seismically active regions (Fig. 1b). Previous studies have suggested that the low-seismicity region experiences creep (Hirahara et al., 2003). To clarify, we have deleted the phrase “where inferred heterogeneous creep (aseismic slip) occurs (Fig. 1b; Hirahara et al., 2003; Ohzono et al., 2011; Mizoguchi et al., 2007; Nakajima et al., 2013; Ito et al., 2006)” and replaced this with (lines 24–26): “..., suggesting pronounced localization

of hypocenters on the fault plane. Geodetic observations have suggested that this low seismicity distribution is linked to fault creep (Hirahara et al., 2003).”

- The seismicity record (Fig. 1b, covering 1998–2010) shows a clear seismic gap, but the manuscript also refers to an historical earthquake in 1858. Providing the magnitude and additional context for this event would be valuable.

Reply) In the revision, we have added the hypocenter of the 1858 Hietsu earthquake to Fig. 1a and a sentence about the main characteristics of this earthquake to the first paragraph of the Introduction (lines 20–21). These revisions more clearly emphasize the contrast between seismogenic and low-seismicity regions along the fault.

- I suggest including information from recent geodetic and seismological studies that quantify fault coupling from, for example, GPS-derived slip rates and spatial variations in creep behavior along the fault system. Adding a new figure summarizing these data (e.g., updated Fig. 1) would help readers visualize the structural and kinematic framework.

Reply) Thank you for your suggestion. We agree that presenting additional structural and kinematic information is essential for placing our results in the proper seismo-tectonic context. In response to this comment, we added a summary of recent GPS and EDM observations (lines 25–33). Previous geodetic observations are consistent with fault creep (Hirahara et al., 2003). In particular, EDM observations (1981–1999) indicate creep at ~ 1.5 mm/yr along the central part of the fault (Fig. 1b; Tada, 1998; GSI, 2000). Later GPS observations (1998–2006) indicate a locked state along the same fault segment, suggesting possible time-dependent behavior (Ohzono et al., 2011). In addition, analyses of InSAR and GNSS data (2007–2010) show that the interseismic strain-rate gradient varies over time (Takada et al., 2018). This reflects temporal changes in seismicity along the central part of the fault, which is associated with local stress variations in the seismogenic zone close to the fault tips (Katsumata et al., 2010). We have updated the hypocenter distributions in Fig. 1b using the time period from 1998 to 2025. We believe these additions will help readers to better understand the significance of the latest seismological and geodetic findings in this region.

In the Discussion, consider exploring whether variations in graphene oxide abundance correlate with differences in slip behavior between fault segments (i.e., creeping vs. seismogenic zones). Since

graphene oxide is thermally unstable above ~ 200 °C, it might be preferentially preserved in (shallow?), creeping segments and absent in (deeper?), seismically active ones, as frictional heating during seismic slip will produce temperature rise well above 200 degrees in the gouges. This relationship could provide an important mechanistic link between carbon chemistry, shear localization, and fault slip behaviour.

Reply) Thank you very much for this insightful comment. The potential relationship between graphene oxide and fault slip behavior is a crucial point in our manuscript. As the reviewer points out, frictional heating during earthquakes may be a key factor in determining the instability of graphene oxide. We have substantially revised lines 308–317 to clarify this point, as well as to expand our discussion comparing the seismically active and low-seismicity regions along the fault. We now discuss whether regional (e.g., the geothermal gradient) and short-lived (e.g., seismic slip on a fault segment) conditions could control the presence of graphene oxide within the fault and whether these are compatible with distinct seismological observations along the central segment of the fault system.

To summarise, this is a novel and promising contribution that identifies graphene oxide as a potential control on fault slip behavior. To maximize its impact, the manuscript would benefit from:

- A clearer, more structured presentation of field and analytical results;
- A more comprehensive and better-integrated discussion of the seismo-tectonic setting; and
- A concise discussion about any potential/inferred link between graphene oxide formation, its thermal/mechanical stability, and the observed creeping vs. seismic behavior of the fault system.

Reply) We thank the reviewer for these constructive and encouraging comments. As noted in our replies above, we have substantially modified sections of the manuscript to address the reviewer's comments. To summarize, we have:

- reorganized the presentation of our results to more clearly describe our sampling locations, fieldwork results, and microstructural observations.
- added and discussed new data acquired from TEM observations.
- expanded our descriptions (and figures) of recent geodetic and seismological datasets relevant to our understanding of the Atotsugawa Fault System.
- strengthened the contextual links between the existing datasets and our new results.
- expanded the discussion section to address specific comments related to the formation and tectonic significance of graphene oxide.

Overall, we believe that these revisions substantially improve the clarity and impact of the manuscript,

and provide better context to evaluate our new findings in the context of a broader regional framework. We are grateful to the reviewer for their valuable suggestions, which greatly helped to strengthen this study.

Reviewer #2 (Remarks to the Author):

The authors present an interesting study on aspects of nanolith geology, specifically the geology of graphene, with implications for the frictional behavior of shear zones located in Japan. The work has merit in its focus on the identification of natural graphene. However, the material identified does not adequately meet the IUPAC standards and formal definitions required to be classified as graphene oxide or reduced graphene oxide, including the absence of a characterization technique capable of determining the material's thickness. Consequently, the authors' main finding is not fully supported by the presented data—though this does not diminish the scientific value of the study.

Reply) Thank you for this constructive and insightful comment. We appreciate the reviewer's careful assessment of our materials based on IUPAC definitions, as well as the overall positive evaluation of our study. As suggested by the reviewer, Raman spectroscopy and X-ray photoelectron spectroscopy do not allow unambiguous determination of whether the material has a strictly single-layer structure that would define graphene oxide. To address this, the revised manuscript (including the methods section and supplementary materials) refers to the material as “graphene oxide–like carbonaceous material”. Our new transmission electron microscopy (TEM) observations (lines 227–255 and Fig. 6) clearly reveal that the analyzed graphene oxide has a single-layer structure. This indicates that the material satisfies the criteria for classification as graphene oxide based on IUPAC standards. We hope that our more careful use of the terms used to describe the graphene material, together with the detail revealed by our new TEM analysis, adequately address the reviewer's concerns.

The frictional considerations and interpretations regarding the behavior of the graphite nanoplatelets (graphene oxide–like or –based) are indeed relevant and should become the main focus of the revised manuscript, including the presentation and assessment of the primary coefficient of friction data identified by the authors, which may lead to meaningful interpretations of fault movement processes with significance for the scientific and academic community.

Reply) Thank you for this constructive comment. As pointed out by the reviewer, the frictional behavior of graphene oxide and related materials represents one of the most important conclusions of our paper. We have revised the manuscript to emphasize this result (lines 54–60). We believe that the revised manuscript better emphasizes the significance of graphene oxide to a broader research field, including structural geology, tribology and seismology. The presence of graphene in shear zones offers a new framework to interpret fault movement processes that are of broad interest to the scientific

community.

Further specific and complementary comments on the manuscript are provided below:

When the authors state “When mixed with quartz, a major mineral in the crust,” it would be advisable to specify that it refers to the continental crust.

Reply) As suggested, we have revised the text in line 42 to refer to the continental crust.

The authors should be cautious with the use of expressions such as “single layers of graphene,” since, strictly speaking, graphene is intrinsically a single-layer structure throughout its definition. Any material composed of more than one layer is, by definition, a graphite nanoplatelet, which may exhibit behaviors analogous to those of graphene or graphene oxide, but should not formally be referred to by these terms.

Reply) We appreciate this clarification regarding the correct terminology used to refer to graphene. In the revised paper, we have carefully considered each usage of this and similar terms, and modified the terms where necessary to avoid mistakes or ambiguity. Following the reviewer’s suggestion, we also conducted new TEM analyses of the graphene materials, confirming that the graphene oxide-like material has a single-layer structure. As mentioned previously, this structure meets the IUPAC definitions for graphene oxide, and we have therefore used the term “graphene” in the revised manuscript (e.g., lines 271–273). In the revised manuscript, we have added detailed information on the new TEM analyses and observations (lines 227–255 and Fig. 6) and incorporated these into the revised Discussion (lines 270–271, 277–278 and 359–361).

Although the caption of Figure 1 explains what is being represented, this does not eliminate the need to include a legend within the figure itself, for two main reasons: (i) some readers may be color-blind or have other conditions that prevent them from accurately distinguishing colors; and (ii) every figure must be entirely self-contained so that a reader can interpret it without any external reference, which requires the inclusion of a caption within the figure. In addition, the meaning of the area outlined with a blue dashed line between X–X’ was not described, not even in the caption.

Reply) The reviewer’s detailed comments on Fig. 1 offered valuable guidance for enhancing the clarity of this figure. We have added a legend to Fig. 1a (on the right side), and the color scheme has been

adjusted (black, orange, and blue) to enhance visual distinction. In Fig. 1b, we added information on the distribution of crustal fluids based on seismic tomography analysis from Nakajima et al. (2010). Shaded patterns (regions of the aqueous fluids and partial melting) were also introduced to clarify the locations of these areas, including the region outlined by the blue dashed line between X–X' (Fig. 1b). Similar color adjustments were applied to Fig. 1c. Following other comments below, corresponding adjustments were also made to Fig. 2.

I did not understand what the authors meant in line 79: “In this paragraph, we summarize geochronological data from each sampling area,” since no geochronological data or crustal evolution based on isotopic data are actually presented. Instead, the paragraph describes the location contexts of the sites where the samples used in the study were collected.

Reply) We apologize for the confusion caused by the inaccurate wording in the original text. We have revised these sentences (lines 99–123) to clarify that the paragraph describes the locations of the sampling sites used in this study.

In line 95, what does the “black material” refer to? Does it also correspond to a CM? It is important to provide a clearer description of this material, which is currently much less well characterized than the other samples.

Reply) In the revised manuscript, we clarify that the black material corresponds to carbonaceous material (CM) in line 124.

The same issue regarding the use of colors and the absence of an internal legend in Figure 1 also occurs in Figure 2. Therefore, the authors will need to revise the way this figure is presented.

Reply) We thank the reviewer for this helpful comment regarding the figures. We have revised Figs. 2e–f by improving the color scheme and adding a clear internal legend to define the polygons.

Personally, I would prefer the Methods section to be placed immediately after the Introduction in the manuscript, since there are several moments in the text—particularly in the “Results” section—where the authors describe sampling and treatment procedures that are only explained at the end of the paper, which disrupts the textual cohesion. Although the authors clearly made an effort to present the main methodological aspects and to transparently provide the remaining details as supplementary materials,

I found the absence of a table compiling all collected samples and their correlation with the evaluated shear zone noticeable. Such a table could be included as part of the supplementary materials.

Reply) We thank the reviewer for this comment. We agree that moving the Methods section to earlier in the manuscript may help the overall flow of text. However, in accordance with Nature Communications guidelines, we have chosen to keep the Methods in a separate section at the end of the paper. To improve readability, we have revised the introduction to clarify which analytical techniques are required for the identification of graphene oxide, and we briefly outline the advantages of each method (lines 61–70). This revision helps to link the introduction and the results, improving the overall coherence of the text. We have also revised a number of specific sections in the manuscript (e.g., lines 152–160 and 186–192) to ensure that it can be read smoothly without requiring frequent reference to the detailed methods. Finally, to improve clarity in the Supplementary Information, we explicitly distinguish fault gouge from host rock in revised Table S1.

The oxidized materials identified by the authors do not fully meet the most rigorous and conservative standards currently accepted for classifying a substance as graphene oxide or reduced graphene oxide. The authors should instead refer to the characterized and diagnosed material as oxidized graphite nanoplatelets and reduced oxidized graphite nanoplatelets. At most, to emphasize the finding, the terms graphene oxide–based or graphene oxide–like could be used. Both the Raman spectroscopy results (which show the persistence of a significant 2D band) and the XPS data (indicating a substantial amount of carbon atoms maintaining sp^3 hybridization) suggest that these structures are not definitively monolayered. Therefore, they cannot presently be accurately referred to as graphene—or as its oxide or reduced oxide forms.

Reply) Thank you for this comment, which is related to the first comment addressed above. We agree with the reviewer that Raman spectroscopy and XPS do not allow determination of whether the carbonaceous material has a single-layer structure. As our analyses cannot reliably confirm that the material occurs as nanoparticles, we are also not confident in applying the term “nanoplatelets”. In the revised manuscript, we have followed the reviewer’s suggestion to refer to the carbonaceous material as “graphene oxide–like carbonaceous material (GOCM)”. In order to better characterize the CM, we conducted new transmission electron microscopy (TEM) analysis to examine the structure and thickness of the material (lines 227–255). The new observations provide direct evidence that the CM occurs as single-layer sheets (Fig. 6 in the revised manuscript), meaning that we can classify the material as single-layer graphene oxide.

At present, all geological substances of this type are considered graphene-like, since the intrinsically

complex nature of metamorphic environments would make it highly unlikely for structures to be preserved that fully satisfy the chemical criteria and formalism required to be classified as "true" graphene or graphene oxide. Therefore, the main finding of the authors' work does not actually reflect the extraordinary nature initially suggested. If the authors had indeed identified a natural substance that fully meets the criteria for classification as graphene oxide, the discovery would be extremely significant and of high impact—but this is not supported by the data presented in the study.

Reply) Thank you for your valuable comments. Following the reviewer's advice, we conducted additional TEM observations to demonstrate that the graphene oxide-like materials in the fault have a strictly single-layer structure (lines 227–255), satisfying the classification criteria for graphene oxide (Bianco et al., 2013; Wick et al. 2014). As noted by the reviewer, the presence of graphene oxide in natural geological samples is an extremely significant finding because of the unique frictional properties of graphene oxide. We believe that the evidence for graphene oxide provided by our new TEM observations highlights the impact and significance of our work, as it represents the first report of graphene oxide in a natural fault zone.

A recommendation to the authors is to conduct a thorough literature review focused on the geology of graphene, in order to accurately present the current state of the art in their study. Although the authors employ analytical techniques that are pertinent and appropriate for the work, these methods alone do not reflect advances in geological science regarding the identification of geological graphene-like structures, particularly those found in metamorphic environments. Some examples of relevant studies that the authors should consult and use to update their literature review include:

https://doi.org/10.1007/978-3-031-04435-9_34

<https://doi.org/10.1038/s41561-025-01803-3>

<https://doi.org/10.1007/s12040-025-02619-w>

<https://doi.org/10.1002/chem.201500116>

<https://doi.org/10.1590/1517-7076-RMAT-2022-0122>

<https://www.taylorfrancis.com/chapters/edit/10.1201/b19679-12/natural-graphene-based-shungite-nanocarbon-natalia-rozhkova-sergey-rozhkov-andrey-goryunov>

<https://doi.org/10.1134/S1028334X20110112>

<https://doi.org/10.1002/jrs.6474>

<https://doi.org/10.3390/c9030075>

<https://doi.org/10.1080/19475411.2014.885913>

Reply) Following the reviewer's recommendations, we conducted a focused literature review on geological studies of graphene and related materials. Among the references suggested by the reviewer, we have cited the recent review paper on geological applications of graphene by Nobre (2025), as well as representative studies on incompletely exfoliated graphene–graphite phases (Mukherjee & Venkatesh, 2025) and a comprehensive review of shungite (Sheka & Rozhkova, 2014). We have revised the Introduction (lines 54–58) to summarize previous geological studies of graphene-related materials. We then emphasize the paucity of geological and seismological studies on the importance of graphene in a tectonic context (lines 58–60). In the revised Discussion (lines 356–359), we incorporate information from these references to discuss the proposed formation mechanisms of graphene. As pointed out by the reviewer, the materials reported in these references should not strictly be referred to as graphene, but rather as graphene-like materials. In contrast, our study provides the first identification of single-layer graphene oxide within a fault. These revisions better clarify the context of our work and highlight the significance of our new findings.

The frictional properties proposed for the graphene oxide–like or –based material identified by the authors appear to be associated with quantum phenomena that are not observable at the bulk scale in graphite. This is a relevant finding and could be further explored as the main focus of the study.

Reply) Thank you for highlighting the significance of the peculiar frictional properties of graphene oxide arising from quantum effects. We carefully considered this perspective and further examined several potentially relevant friction mechanisms, as outlined below. In pure graphene, superlubricity has been attributed to moiré patterns generated by rotational misalignment between stacked graphene sheets (Dienwiebel et al., 2004). Such effects arise from direct graphene–graphene contact. While moiré patterns have been shown to play a crucial role in achieving superlubricity, the long-standing challenge of achieving superlubricity has been attributed to surface roughness, which tends to destroy moiré patterns (Bai et al., 2023). Graphene oxide is characterized by abundant oxygen-containing

functional groups, such as hydroxyl and epoxy groups, bound to the basal plane (Dreyer et al., 2010; He et al., 1998). The presence of oxygen-containing functional groups disrupts the structure necessary for moiré-pattern-induced superlubricity, making this mechanism improbable for graphene oxide. In this study, we interpret the frictional behavior of graphene oxide to be controlled by a different mechanism, in which interactions between oxygen-containing functional groups on graphene sheets and interlayer water promote ultralow friction (Liang et al., 2019). In addition, experimental and theoretical studies of graphene have shown that, under increasing load, the sliding potential energy surface can evolve from a corrugated to a flattened landscape (Sun et al., 2018). As a result, friction collapse occurs, leading to a friction coefficient of 0. However, such a transition has been reported to require extremely high normal stresses (>100 GPa) even in idealized graphene. The presence of functional groups and electrostatic repulsion in graphene oxide is expected to substantially modify this behavior. The potential link between the frictional properties of graphene oxide and quantum phenomena remains an important subject for future investigation, but it lies beyond the scope of the present study. Such studies have the potential to refine and extend the frictional mechanisms proposed in this work.

To support the estimation of the thickness of the analyzed nanoplatelets, it would be advisable to perform a transmission electron microscopy (TEM) or atomic force microscopy (AFM) analysis, which would allow for a more accurate determination of the thickness of the discovered material.

Reply) This suggestion motivated us to perform TEM analysis of our fault gouge samples (lines 227–255). As described in the revised text and shown in the new Fig. 6, the TEM observations provide critical insights into the nanostructure of the graphene oxide. Specifically, the graphene oxide occurs as a strict single-layer structure that can be oriented in various directions. These new findings further highlight the impact and significance of our work by confirming that the carbonaceous material studied here can indeed be classified as graphene oxide.

One question that does not seem to have been sufficiently discussed or clarified by the authors—and which is relevant to the overall discussion—is whether the amorphous carbonaceous precursor material crystallized directly to form the graphene oxide-like or -based nanoplatelets exhibiting behavior analogous to graphene oxide, or whether a thicker (bulk) graphitic material first crystallized and was subsequently cleaved and oxidized during shear deformation. This distinction may have important implications for structural geology and for the characterization of the properties of nanoliths.

Reply) Thank you very much for this important comment regarding the possible formation mechanisms of graphene materials. We have revised the discussion section to address this comment (lines 347–363). Specifically, we now discuss the possibility that two distinct processes may explain the formation of graphene oxide in active faults: (1) Amorphous carbon derived from organic matter transforms into graphene oxide through graphitization during fault slip (Sheka & Rozhkova, 2014; Nobre, 2025). (2) Crystalline graphite undergoes exfoliation and oxidation during shear deformation, producing graphene oxide (Mukherjee & Venkatesh, 2025; Sheka & Rozhkova, 2014). Both of these processes are feasible under the environmental conditions found in the studied fault zone, because the sedimentary (CM-bearing) host rocks of the Tetori Group are widely distributed and fluids are supplied along the fault zones from depth.

Although we recognize that distinguishing between these two mechanisms will require further analysis, we have revised the discussion (lines 355–363) to interpret our preferred mechanism: that graphene oxide was more likely formed through the transformation of amorphous carbon derived from organic matter by tribochemical reactions during fault friction. Extensive exfoliation studies in materials science (Stankovich et al., 2007; Sumdani et al., 2021), together with the analysis of graphene/graphite in geological samples (Sheka & Rozhkova, 2014), suggest that complete exfoliation of graphite by mechanical and chemical processes is difficult to achieve. However, our TEM observations show that graphene oxide within the studied fault zone lacks any discernible stacking structure and occurs as single sheets with lateral dimensions of 3–10 nm. On this basis, it is conceivable that the graphene oxide found in this work formed directly from amorphous carbon, and that tribochemical process can promote the enlargement of the graphene oxide sheet size. This interpretation is consistent with graphite enrichment models involving pressure-solution previously proposed for the Atotsugawa Fault System by Oohashi et al. (2012). Pressure-solution or diffusive mass transfer can preferentially remove water-soluble minerals such as quartz and carbonates from rocks, leading to an enrichment in relatively insoluble phases such as carbon. Although the revised text introduces these two potential mechanisms and discusses our preferred model, further work will be required to determine the origin of the graphene in detail.

Tracking number: NCOMMS-25-45653A

Dear Editor and reviewers,

We thank the reviewers for their comments, which have improved the overall clarity and quality of our work. Below, we provide our point-by-point responses. In this document and in the revised manuscript, our comments to Reviewer #1 are shown in blue and our replies to Reviewer #2 are shown in orange.

Should you have any further comments or questions, we would be pleased to address them.

Reviewer #1 (Remarks to the Author):

Dear Editor,

I have now gone through the revised version of the manuscript and figures, and I find that the paper has improved significantly from the previous version I looked at.

The structure of the manuscript, the logical presentation and discussion of the arguments and the clarity of the data and observations presented in the figures are of an appropriate standard for a journal like Nature Communications.

The Authors have thoroughly addressed my previous comments and criticisms, and took action to improve the text and figures accordingly.

I do not have any further comments or suggestions on the paper and its scientific merit at this stage of the peer review process.

Regards,

Nicola De Paola

Reply) We sincerely thank the reviewer for the careful evaluation of our revised manuscript and for the positive and encouraging comments. We are pleased that the reviewer finds the manuscript significantly improved and considers its structure, presentation, and clarity appropriate for publication

in Nature Communications.

Reviewer #2 (Remarks to the Author):

The authors demonstrated a strong commitment to addressing each point and question raised during the first round of review, which is commendable. The manuscript has improved in quality as a result of these efforts. Additional and complementary comments are presented below:

In line 42, the expression “a major mineral in the continental crust” is unnecessary. It would be far more relevant to the discussion to clearly present the principal and accessory mineralogy, and ideally the mineral paragenesis. Is the rock composed exclusively of graphite and quartz? If so, the absence of phyllosilicates or other platy minerals would make the systematic cleavage behavior of the material even more intriguing and worthy of deeper discussion. The presence of platy minerals is well known to play a fundamental role in processes aimed at exfoliating graphite into graphene-like structures. This aspect requires further elaboration, particularly in the context of top-down approaches for the production of graphene-like materials. It is essential to clarify whether the fundamental process responsible for the formation of these crystalline structures should be interpreted as top-down or bottom-up, as this distinction has significant implications for the scientific interpretation and technological relevance of the study.

Reply) Thank you very much for this insightful comment. We have removed the expression “a major mineral in the continental crust” from line 42. As the reviewer points out, XRD analysis of the fault gouge sample (AFST) identified quartz, illite, and muscovite based on multiple diffraction peaks (Fig. S9). A peak at $2\theta = 12.4^\circ$ was interpreted to reflect contributions from both graphene oxide and chlorite. The XRD and TEM suggest that graphene oxide and clay minerals may be intermixed within the fault gouge (Figs. S9, S11–12). Previous studies in materials science have demonstrated that platy minerals interact with graphene and contribute to its formation. Clay minerals readily adsorb graphene oxide and can yield graphene-like materials upon thermal treatment (Nethravathi et al., 2008 *Carbon*), while mica can act as a substrate for graphene, facilitating sliding and epitaxial growth (Lippert et al., 2013 *Carbon*). In our study, we report the occurrence of single-layer graphene oxide nanosheets in natural fault rocks. We propose that tribochemical reactions during fault slip can induce the transformation of amorphous carbon into graphene oxide. Thus, interactions with clay minerals may influence graphene oxide growth and frictional behavior, suggesting a bottom-up approach involving tribochemical reactions and mineral surfaces, rather than top-down approach such as exfoliation of graphite. We acknowledge that the detailed mechanisms, including the contribution of platy minerals, require further experimental studies. Accordingly, we have revised lines 122–124 and 372–376 to

include a discussion of top-down and bottom-up approaches to the formation of graphene oxide in faults, and to incorporate previous studies on interactions with platy minerals.

The authors report the presence of illite in the regions where nano-scale structures analogous to graphene oxide were identified. It is essential to characterize the micro- to nano-structural and textural relationships between the carbonaceous material and the clay mineral, as this association may be critical for constraining the genetic interpretation of the processes responsible for the formation of these crystalline structures. A more detailed analysis of these relationships would substantially strengthen the discussion and enhance the potential scientific impact of the manuscript within the broader research community.

Reply) Thank you very much for this insightful comment regarding the nano-scale relationships between graphene oxide and clay minerals. As shown in yellow dashed region in Fig. 6 and Fig. S11, the interface between illite and carbonaceous material has a roughness at the nanoscale. A SAED pattern collected from location 9 in the TEM image is consistent with graphene, whereas that from location 19 shows diffraction patterns attributable to both graphene and clay minerals (Fig. S12). These observations imply that graphene oxide and illite may be locally mixed. We suggest that some graphene oxide nanosheets can be arranged along the irregular interfaces with clay minerals, potentially resulting in variable orientations. In our HRTEM observations, it was challenging to examine the detailed interface structures between clay minerals and graphene oxide. To this end, lattice fringes arising from diffraction contrast at the boundaries between graphene and clay minerals must be more carefully identified. However, this is beyond the scope of the original target of the present study (first report of graphene oxide in fault materials) and we would welcome to future research on this important topics. To address the reviewer's suggestion, we have revised the manuscript to include additional description of the observed nano-scale textural relationships in the TEM results section in lines 248–252.

Overall, the revisions implemented by the authors are relevant and contribute positively to improving the discussion. The remaining task is to address the points raised above in order to better focus and centralize the genetic discussion of the nano-structures.

Reply) As described above, we have revised the manuscript to include additional discussion on the nano-scale structure of graphene oxide based on the reviewer's comments. We sincerely thank the reviewer for these valuable suggestions, which have strengthened this study.